



# Stream temperature evolution in Switzerland over the last 50 years

Adrien Michel[1,2], Tristan Brauchli[1,3,4], Michael Lehning[1,2], Bettina Schaefli[3], and Hendrik Huwald[1,2]

[1]School of Architecture, Civil and Environmental Engineering, École Polytechnique Fédérale de Lausanne (EPFL), Switzerland
[2]WSL Institute for Snow and Avalanche Research (SLF), Davos, Switzerland
[3]Faculty of Geosciences and Environment, University of Lausanne, Lausanne, Switzerland
[4]Centre de Recherche sur l'Environnement Alpin (CREALP), Sion, Switzerland

**Correspondence:** A. Michel (adrien.michel@epfl.ch)

**Abstract.** Stream temperature is a key hydrological variable for ecosystem and water resources management and is particularly sensitive to climate warming. Despite the wealth of meteorological and hydrological data, few studies have quantified observed stream temperature trends in the Alps. This study presents a detailed analysis of stream temperatures in 52 catchments in Switzerland, a country covering a wide range of alpine and lowland hydrological regimes. The influence of discharge, precipitation, air temperature and upstream lakes on stream temperatures and their temporal trends is analysed from multi-decade to seasonal time scales. Stream temperature has significantly increased over the past 5 decades, with positive trends for all four seasons. The mean trends for the last 20 years are +0.37°C per decade for water temperature, resulting from joint effects of trends in air temperature (+0.39°C per decade) in discharge (-10.1% per decade) and in precipitation (-9.3% per decade). For a longer time period (1979-2018), the trends are +0.33°C per decade for water temperature, +0.46°C per decade for air temperature, -3.0% per decade for discharge and -1.3% per decade for precipitation. We furthermore show that in alpine streams, snow and glacier melt compensates air temperature warming trends in a transient way. Lakes, on the contrary have a strengthening effect on downstream water temperature trends at all elevations. The identified stream temperature trends are furthermore shown to have critical impacts on ecological temperature thresholds, especially in lowland rivers, suggesting that these are becoming more vulnerable to the increasing air temperature forcing. Resilient alpine rivers are expected to become more vulnerable to warming in the near future due to the expected reductions in snow- and glacier melt inputs.

## 1 Introduction

Water temperature is recognized as a key variable for assessing water quality of freshwater ecosystems in streams and lakes (Poole and Berman, 2001). It influences the metabolic activity of aquatic organisms but also biochemical cycles (e.g. dissolved oxygen, carbon fluxes) of such environments (Stumm and Morgan, 1996; Yvon-Durocher et al., 2010). Water temperature is a key factor for many industrial sectors too, e.g. as cooling water for electricity production or in large buildings. Water temperature also strongly influences the quality of drinking water by modifying its biochemical properties (Delpla et al., 2009).





The ongoing climate change could drastically modify this fragile balance by altering the energy balance and by reducing water availability during warm and dry months of the year. At global scale, several studies have shown a clear trend during the last decades in lake surface temperature (Dokulil, 2014; O'Reilly et al., 2015) and in stream temperature at various locations (Morrison et al., 2002; Hari et al., 2006; Webb and Nobilis, 2007; Hannah and Garner, 2015; Watts et al., 2015).

In the last 50 years, a general warming trend has been observed in Swiss rivers (FOEN, 2012) with a singularity in 1987/1988: an abrupt step change of about $+1°C$ (Hari et al., 2006; FOEN, 2012). This corresponds to the global regime shift observed at the same period (Reid et al., 2016). This warming is more pronounced in winter, spring and summer than in autumn (North et al., 2013). For the period 1972 to 2001, no general trend is observed before or after the abrupt 1987/1988 warming (Hari et al., 2006). However, for some rivers, a clear trend exists in addition to the 1987/1988 shift. For example, the Rhine river

in Basel shows an increase of about $3°C$ between 1960 and 2010 (FOEN, 2012), and for rivers feeding into Lake Lugano, an increase between $1.5$ and $4.3°C$ has been observed for the period 1979-2012 (Lepori et al., 2015). The 1987/1988 shift is also observed in groundwater temperature, but more attenuated in time than detected in surface water temperature (Figura et al., 2011). The main driver of the observed river warming in Switzerland is air temperature, with the 1987/1988 increase due to the shift in North Atlantic Oscillation and Atlantic Multi-decadal Oscillation indices (Hari et al., 2006; Figura et al., 2011; Lepori

et al., 2015). However, urbanization is also considered as an additional driver in some catchments due to the increasing fraction of sealed surfaces absorbing more radiative energy than natural surfaces and transferring this heat to surface runoff (Lepori et al., 2015).

    From a general perspective, the main proxy for water temperature is air temperature, with a clear non-linear relationship at sub-yearly scale (such relationships often show typical seasonal hysteresis; Morrill et al. (2005)), but a linear relationship on

longer time scales (Lepori et al., 2015). The heat flux at the water surface is composed of the solar radiation, the net longwave radiation, the latent heat flux and the sensible heat transfers. Studies have shown that the main components of the total energy budget are the radiative components (Caissie, 2006; Webb et al., 2008). Friction at the stream bed and stream bed/water heat exchanges have been shown to be non-negligible components in some cases, e.g. steep slopes and altitudinal gradients (Webb and Zhang, 1997; Moore et al., 2005; Caissie, 2006; Küry et al., 2017). These heat exchanges are more important in the total

heat budget in autumn when residual heat from the summer is still stored in the ground and when riparian vegetation is present. In the latter case, induced shading and reduced wind velocity decrease surface turbulent heat fluxes.

    Groundwater temperature is also an important factor, especially close to the river source (Caissie, 2006) or in areas of significant groundwater infiltration. In Switzerland, this is especially important for high alpine rivers, which are mainly fed by glacier or snow melt, and thus sensitive to changes in the amount of melting and in seasonality (Harrington et al., 2017; Küry

et al., 2017). Discharge is an important driver of water temperature; at different stream flow stages, different water sources (soil water, groundwater, overland flow) are contributing to the total discharge. Streamflow volume directly influences the heat balance as the wetted perimeter of the river modifies atmospheric and ground heat exchanges (Caissie, 2006; Webb and Nobilis, 2007; Toffolon and Piccolroaz, 2015) and the volume influences the temperature change for a given amount of heat exchanged. Accordingly, discharge influences water temperature in a potentially highly non-linear way. This explains partly why many





statistics-based water temperature models do require discharge as an explanatory variable (Gallice et al., 2016; Toffolon and Piccolroaz, 2015).

Anthropogenic influences on stream temperature have been observed due to urbanization and channelization (Webb, 1996; Lepori et al., 2015), vegetation removal (Johnson and Jones, 2000; Moore et al., 2005), use of water for industrial cooling (Webb, 1996; Råman Vinnå et al., 2018) or intake for irrigation agriculture (Caissie, 2006). Hydro-peaking (sudden release of water at sub-daily time scale from hydropower plants) and related thermopeaking have been shown to reduce the impact of summer heat waves on stream temperature (Feng et al., 2018), accompanied, however, with so far relatively poorly known effects on aquatic life (Zolezzi et al., 2011). Overall, most human influences have been proven to modify the relationship between air and water temperature, leading to a weaker correlation (Webb et al., 2008).

In this paper, we investigate the evolution of stream temperature in Switzerland for 52 catchments since the beginning of automatic measurement networks in the 1960s covering a variety of landscapes from high alpine to lowland hydrological systems. Analysis is carried out on raw data for the whole time period, 1963-2018 for the longer ones, and a linear regression analysis is performed over two periods, 1979-2018 and 1999-2018. Trends in water temperature, along with trends in discharge, air temperature and precipitation are analysed using de-seasonalized time series. The 1987/1988 water temperature shift described in the literature (Hari et al., 2006; Figura et al., 2011; North et al., 2013) is discussed in the context of extended historical time series. Given the variety of fluvial regimes (alpine, low-land, disturbed) found in Switzerland, sensitivity of water temperature change to this parameter is also examined. Sensitivity to other topographical characteristics such as the mean catchment elevation and surface area as well as the fraction of glacier coverage are also investigated. The analysis is done at yearly scale and at seasonal scale. In spite of the availability of the data sets, they have not been analysed until now at such scale (52 catchments) and at sub-yearly resolution in the context of climate change, especially with the focus on the response of the different hydrological regimes. In addition, the effect of lakes on river water temperature and the memory effect in the hydrological system (influence from season to season) are studied. Various effects including snow melt, glacier retreat or influence of lakes are also discussed and some relevant indicators for Switzerland are presented.

This study develops the first comprehensive analysis of stream temperature and related variables in Switzerland identifying changes up to date and providing a reference for gauging future evolution and scenarios in view of ongoing climate change.

## 2 Description of data

### 2.1 Stream temperature and discharge data

Water temperature and discharge data along with physiographic characteristics are provided by the Swiss Federal Office for the Environment (FOEN, 2019), by the Office for water and waste of the Canton of Bern (AWA, 2019) and by the Office for waste, water, energy and air of the Canton Zurich (AWEL, 2019). The discharge and water temperature data from FOEN are provided at daily time step, while the AWA and AWEL water data are provided at hourly time step. For most of the FOEN stations, hourly data also exist (see Table 1). While discharge measurements exist for some stations since the beginning of the 20[th] century (mainly installed in the context of hydropower infrastructure projects), water temperature records appeared only





in the 1960's. In the present study, stations with sufficiently long times series of water temperature and discharge are selected (observations available from before 1980 for FOEN data and before 2000 for AWA and AWEL data). Some stations fulfilling a priori these conditions have been removed for other reasons that are detailed in Table S1 in supplementary. Data from other Swiss Cantons have been investigated, but to the best of our knowledge, no other Swiss Canton provides water temperature
measurements before 2000. In particular, no data from the Canton of Ticino could be used, so only one catchment is located on the southern side of the Alps in this study. Note that a recent study already discussed the river warming in Ticino (Lepori et al., 2015).

The 52 selected watersheds, presented in Table 1 and Figure 1, cover a wide range of catchment areas (from a few km$^2$ to tens of thousands km$^2$) and mean elevations (from 450 m to more than 2500 m). Due to the complex topography of the
country, the partitioning between solid and liquid precipitation can strongly vary over small distances. Combined with the presence of glaciers in some catchments, this factor naturally influences the hydrological response characterized through the hydrological regime (Aschwanden et al., 1985). The selected catchments are representative of all natural hydrological regimes found in Switzerland except southern Alps regimes; they can also be influenced by human activities (hydropower production, lake regulation, water intake or release). The basins are classified into four different categories (Piccolroaz et al., 2016):

– **Swiss Plateau and Jura regime (SPJ)**: on the lower part of the country, most of the precipitation falls as rain. The hydrological response is driven by precipitation and evapotranspiration. The annual cycle in discharge is moderate with a minimum in summer and exhibits a high inter-annual variability depending on regional precipitation patterns.

– **Alpine regime (ALP)**: at higher elevations, both the discharge and thermal regimes are strongly influenced by snow and glacier melt. A pronounced annual cycle is identifiable, with a maximum between March and July depending on the
mean basin elevation and on the fraction of glacier coverage, and a minimum during the winter season.

– **Downstream lake regime (DLA)**: Switzerland has many large lakes, most of them being regulated for flood control purposes (with the notable exception of Lake Constance). As a result, downstream rivers are not only influenced by the lake itself (natural buffer) but also by its anthropogenic management (extra smoothing).

– **Regime strongly influenced by hydropeaking (HYP)**: roughly 55% of Switzerland's electricity production stems from
hydropower plants (Schaefli et al., 2019). Storage facilities at high elevation impact the hydrological regime in the lowlands by controlled intermittent release of large volumes of cold water.

## 2.2 Meteorological data

To each hydrometric gauging station, one or more meteorological stations, operated by the Federal Office of Meteorology and Climatology, MeteoSwiss, have been associated (IDAWEB, 2019). These stations were selected according to proximity of the
catchments in order to be representative of the local meteorological conditions. Only stations with sufficiently long data records at daily time scale were kept.





**Table 1.** Physiographic characteristics and data availability for water temperature and discharge of the 52 selected catchments. The IDs are the ones used by the data providers and the ones with an asterix represent stations where no hourly temperature measurements are available. The providers are the Swiss Federal Office for the Environment (FOEN), the Office for water and waste of the Canton of Bern (AWA) and the Office for waste, water, energy and air of the Canton Zurich (AWEL). For each basin, the selected representative MeteoSwiss meteorological stations are indicated. The details of abbreviations of the MeteoSwiss stations can be found in Table S2 in supplementary.

| ID | River | Abbreviation | Temperature measurement | Discharge measurement | Area [km$^2$] | Mean basin elevation [m] | Glacier surface [%] | Hydrological regime | Data provider | Meteorological station |
|---|---|---|---|---|---|---|---|---|---|---|
| 527 | Aabach in Mönchaltorf | Aab-Mon | 1992-2018 | 1992-2018 | 46 | 523 | 0 | SPJ | AWEL | SMA |
| 2135 | Aare in Bern | Aar-Ber | 1971-2018 | 1918-2018 | 2941 | 1596 | 5.8 | DLA | FOEN | BER, INT |
| 2019 | Aare in Brienzwiler | Aar-Bri | 1971-2018 | 1905-2018 | 555 | 2135 | 15.5 | HYP | FOEN | GRH, MER |
| 2016 | Aare in Brugg | Aar-Bru | 1963-2018 | 1916-2018 | 11681 | 1000 | 1.5 | DLA | FOEN | WYN, SMA |
| 2029 | Aare in Brügg-Aegerten | Aar-bra | 1963-2018 | 1989-2016 | 8249 | 1142 | 2.1 | DLA | FOEN | BER, MUB, PAY, NEU |
| 2085 | Aare in Hagneck | Aar-hag | 1971-2018 | 1984-2018 | 5112 | 1368 | 3.4 | DLA | FOEN | BER, MUB |
| 2457 | Aare in Ringgenberg | Aar-Rin | 1964-2018 | 1931-2016 | 1138 | 1951 | 12.1 | DLA | FOEN | MER, INT |
| 2030 | Aare in Thun | Aar-Thu | 1971-2018 | 1906-2018 | 2459 | 1746 | 6.9 | DLA | FOEN | MER, INT |
| A019 | Alte Aare in Lyss | Aar-Lys | 1997-2018 | 1997-2018 | 13 | 462 | 0 | SPJ | AWA | MUB, BER |
| 2170 | Arve in Geneva | Arv-Gva | 1969-2018 | 1904-2018 | 1973 | 1370 | 5 | ALP | FOEN | GVE |
| 2106 | Birs in Münchenstein | Bir-Muc | 1972-2018 | 1917-2018 | 887 | 728 | 0 | SPJ | FOEN | BAS, DEM |
| 2034 | Broye in Payerne | Bro-Pay | 1976-2018 | 1920-2018 | 416 | 715 | 0 | SPJ | FOEN | PAY |
| A062 | Chrouchtalbach in Krauchthal | Chr-Kra | 1999-2018 | 1999-2018 | 16 | 702 | 0 | SPJ | AWA | BER |
| 2070 | Emme in Emmenmatt | Emm-Emm | 1976-2018 | 1974-2018 | 443 | 1065 | 0 | SPJ | FOEN | LAG, NAP |
| 2481 | Engelberger Aa in Buochs | Eaa-Buo | 1983-2018 | 1916-2018 | 228 | 1609 | 2.5 | HYP | FOEN | ENG |
| 522 | Eulach in Winterthur | Eul-Win | 1993-2018 | 1993-2018 | 64 | 541 | 0 | SPJ | AWEL | SMA, TAE |
| 2415 | Glatt in Rheinfelden | Gla-Rhe | 1977-2018 | 1976-2018 | 417 | 503 | 0 | SPJ | FOEN | SMA, KLO |
| 534 | Glatt in Rümlang | Gla-Rum | 1992-2018 | 1992-2018 | 302 | 520 | 0 | DLA | AWEL | SMA |
| 531 | Glatt in Wuhrbrücke | Gla-Wuh | 1993-2018 | 1993-2018 | 64 | 621 | 0 | SPJ | AWEL | SMA |
| 2462 | Inn in S-Chanf | Inn-Sch | 1981-2018 | 1999-2018 | 616 | 2463 | 6.1 | ALP | FOEN | SAM, SIA, BEH |
| A017 | Kander in Frutigen | Kan-Fru | 1995-2018 | 1992-2018 | 180 | 2156 | 14 | ALP | AWA | ABO |
| 517 | Kempt in Illnau | Kem-Ill | 1992-2018 | 1992-2018 | 37 | 615 | 0 | SPJ | AWEL | SMA, TAE |
| 2634* | Kleine Emme in Emmen | Kem-Emm | 1973-2018 | 1936-2018 | 478 | 1054 | 0 | SPJ | FOEN | LUZ, NAP |
| A025 | Langete in Roggwil | Lan-Rog | 1996-2018 | 1996-2018 | 130 | 689 | 0 | SPJ | AWA | KOP, WYN |
| 2243 | Limmat in Baden | Lim-Bad | 1969-2018 | 1951-2018 | 2384 | 1131 | 0.7 | DLA | FOEN | SMA, WAE |
| 2372 | Linth in Mollis | Lin-Mol | 1964-2018 | 1914-2018 | 600 | 1743 | 2.9 | HYP | FOEN | ELM, GLA |
| 2104 | Linth in Weesen | Lin-Wee | 1964-2018 | 1907-2018 | 1062 | 1584 | 1.6 | DLA | FOEN | ELM, GLA RAG |
| 2269 | Lonza in Blatten | Lon-Bla | 1967-2018 | 1956-2018 | 77 | 2624 | 24.7 | ALP | FOEN | ABO, GRH |
| A070 | Luterbach in Oberburg | Lut-Obe | 1994-2018 | 1994-2018 | 34 | 700 | 0 | SPJ | AWA | BER |
| 2109 | Lütschine in Gsteig | Lus-Gst | 1964-2018 | 1908-2018 | 381 | 2050 | 13.5 | ALP | FOEN | INT, GRH |
| 2084 | Muota in Ingenbohl | Muo-Ing | 1974-2018 | 1917-2018 | 317 | 1363 | 0 | HYP | FOEN | ALT |
| A029 | Önz in Heimenhausen | Onz-Hei | 1994-2018 | 1995-2018 | 86 | 582 | 0 | SPJ | AWA | KOP, WYN |
| A031 | Ösch in Koppigen | Osc-Kop | 1997-2018 | 1997-2018 | 39 | 559 | 0 | SPJ | AWA | KOP |
| A049 | Raus in Moutier | Rau-Mou | 1997-2018 | 1997-2018 | 41 | 896 | 0 | SPJ | AWA | DEM |
| 572 | Reppisch in Dietikon | Rep-Die | 1993-2018 | 1993-2018 | 69 | 594 | 0 | DLA | AWEL | SMA |
| 2152 | Reuss in Luzern | Reu-Luz | 1973-2018 | 1922-2018 | 2254 | 1504 | 2.8 | DLA | FOEN | LUZ |
| 2018 | Reuss in Mellingen | Reu-mel | 1969-2018 | 1904-2018 | 3386 | 1259 | 1.8 | DLA | FOEN | LUZ, SMA |
| 2056 | Reuss in Seedorf | Reu-See | 1971-2018 | 1904-2018 | 833 | 2013 | 6.4 | HYP | FOEN | ALT |
| 2473 | Rhein in Diepoldsau | Rhe-Die | 1970-2018 | 1919-2018 | 6299 | 1771 | 0.7 | HYP | FOEN | CHU, RAG, VAD |
| 2143 | Rhein in Rekingen | Rhe-Rek | 1969-2018 | 1904-2018 | 14767 | 1131 | 0.4 | DLA | FOEN | HLL, KLO |
| 2091* | Rhein in Rheinfelden | Rhe-Rhe | 1971-2018 | 1933-2018 | 34524 | 1068 | 1.1 | DLA | FOEN | BAS, KLO |
| 2174 | Rhône in Chancy | Rho-Cha | 1971-2017 | 1904-2017 | 10308 | 1569 | 8.3 | DLA | FOEN | GVE |
| 2009 | Rhône in Porte du Scex | Rho-Pds | 1968-2018 | 1905-2018 | 5238 | 2127 | 11.1 | HYP | FOEN | SIO, GSB |
| 2011* | Rhône in Sion | Rho-Sio | 1974-2018 | 1916-2018 | 3372 | 2291 | 14.2 | HYP | FOEN | SIO, GRC, GRH |
| A047 | Sagibach in Worben | Sag-Wor | 1996-2018 | 1996-2018 | 13 | 459 | 0 | SPJ | AWA | MUB, BER |
| 547 | Sihl in Blattwag | Sih-Bla | 1992-2018 | 1992-2018 | 102 | 1168 | 0 | DLA | AWEL | WAE, EIN |
| A022 | Suze in Villeret | Suz-Vil | 1995-2018 | 1995-2018 | 61 | 1080 | 0 | SPJ | AWA | CDF, CHA |
| 2044 | Thur in Andelfingen | Thu-And | 1963-2018 | 1904-2018 | 1702 | 770 | 0 | SPJ | FOEN | KLO, SAE, STG |
| 2068 | Ticino in Riazzino | Tic-Ria | 1978-2017 | 1997-2018 | 1613 | 1643 | 0.1 | HYP | FOEN | SBE, OTL |
| 570 | Töss in Freienstein | Tos-Fre | 1992-2018 | 1992-2018 | 399 | 626 | 0 | SPJ | AWEL | SMA, TAE |
| 520 | Töss in Ramismuhle | Tos-Ram | 1992-2018 | 1992-2018 | 127 | 803 | 0 | SPJ | AWEL | SMA, TAE |
| 2500 | Worble in Ittigen | Wor-Itt | 1989-2018 | 1989-2018 | 67 | 666 | 0 | SPJ | FOEN | BER |





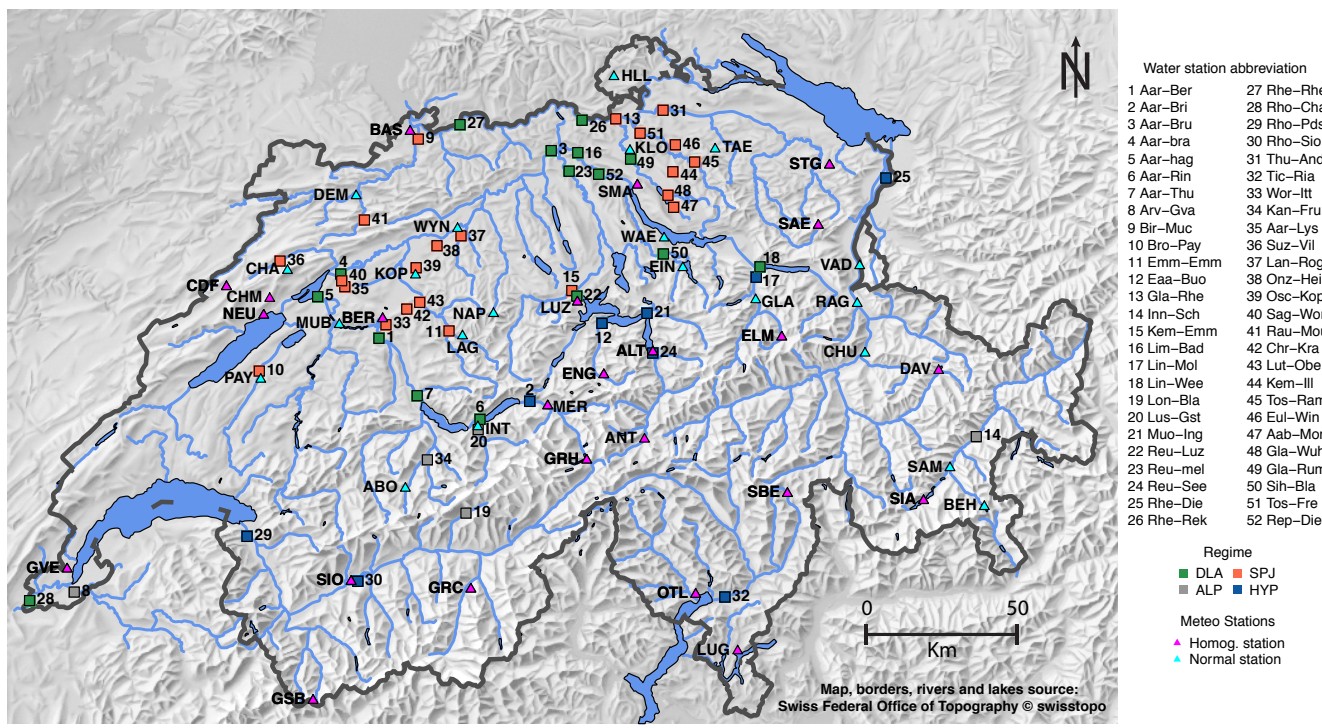

**Figure 1.** Map of Switzerland with the selected hydrometric gauging stations and associated meteorological stations. Abbreviations for hydrometric gauging stations are defined in Table 1 and for meteorological stations in Table S2 in supplementary.

Daily measurements of air temperature and precipitation were compiled and homogeneous time series (Füllemann et al., 2011) were used whenever available. Homogenization done by MeteoSwiss consists of adjusting historic measured values to current measuring standards. Figure 1 shows a map with all sites of water temperature measurement and associated MeteoSwiss stations. In total, 41 MeteoSwiss stations are associated with one or several catchments. Details on the stations are given in
5 Table S2.

### 2.3 Snow water equivalent and glaciers mass balance

Monthly snow water equivalent maps of Switzerland are used as proxy for snow melt. These maps are provided by the WSL Institute for Snow and Avalanche Research, SLF. They are generated using a temperature-index model in which observational SWE data are assimilated with an ensemble Kalman filter (Magnusson et al., 2014; Griessinger et al., 2016). Glacier annual and
10 seasonal (summer and winter) mean local mass balance and surface extent are available for selected glaciers from the GLAMOS data set (GLAMOS, 2018). The mass balance is estimated based on in-situ bi-annual measurements and then extrapolated to the whole glacier area using distributed modelling and point measurements homogenization techniques (Huss et al., 2015) to





retrieve the mean local mass balance. The total mass balance is obtained in this study by multiplying the mean annual and seasonal mass balance (in mm water equivalent per year) by the glacier area.

## 3   Methods

### 3.1   Data pre-processing procedure

In the analysis below, only complete calendar years are retained; sparse or missing data are allowed as long as gaps do not exceed 2 weeks. In daily averaging, missing data are propagated (i.e. one missing data during a day results in a missing day), but they are ignored for seasonal and annual averaging. Seasonal and annual time series are used for all inter-annual comparisons and for inter-variable correlation studies. Daily time series are used for the trend analysis. Indeed, more points are available in daily values than in annual ones, leading to more robust trends (see Section 3.3).

### 3.2   Seasonal signal removal

Before applying linear regression to daily data, the seasonal signal is removed with a method called Seasonal-Trend decomposition based on 'Loess' (STL) (Cleveland et al., 1990), where Loess stands for locally weighted regression (Cleveland and Devlin, 1988; Cleveland et al., 1988). This method is robust with respect to outliers in the time series, able to cope with missing data and with any seasonal signal shape, and is computationally efficient. In addition, the seasonality is allowed to change over

time and this rate of change is parameterized by the user. The STL method has been widely used in other fields, examples of application in hydrology include the work of Hari et al. (2006), Figura et al. (2011) or Humphrey et al. (2016).

Here only the main ideas of the method are presented, full details are given in Cleveland et al. (1990). The Loess fitting method is a local fitting with weights applied to the points that are fitted. The fitting can be locally-linear or locally-quadratic, here we use the locally-linear fitting as in the paper of Cleveland et al. (1990). For any $x_i$ in the neighbourhood of $x$, the Loess,

or the weight applied to the points before doing a local fitting, is defined as:

$$v(x) = W\left(\frac{|x_i - x|}{\lambda_q(x)}\right) \tag{1}$$

Note that $x_i$ is the position of the point, not its value. $\lambda_q(x)$ is defined as the distance to the $q_{th}$ furthest point, $q$ being a parameter of the model discussed below. $W(x)$ (Cleveland and Devlin, 1988; Cleveland et al., 1988) is defined by the tricubic function:

$$W(x) = \begin{cases} (1 - x^3)^3 & \text{for } 0 \leqslant x < 1 \\ 0, & \text{otherwise} \end{cases} \tag{2}$$

So $W(x)$ is large for $x_i$ close to $x$ and becomes zero for $x_i$ further than the $q_{th}$ farthest point. We can see that $q$ will act as a smoothing parameter on the fit obtained with this method.





In the STL algorithm, vectors of data $Y$ are decomposed as follows:

$$Y_i = T_i + S_i + R_i \tag{3}$$

where $T_i$ is the trend term, $S_i$ the seasonal term and $R_i$ the residual term. The algorithm is composed of two iterative loops,
called inner and outer loops. In the inner loop, the time series is first de-trended: $T_i$ is extracted and smoothed with a Loess

fitting as explained above. Then, the seasonal component is extracted with a low-pass filter, and the remaining time series
is again smoothed by Loess. This process is repeated iteratively and encapsulated in an outer loop. In this second loop, the
residuals are analysed and a weight, which is low for outliers, is attributed. These weights are used for the Loess fitting in the
next round of the inner loop. At the end of the loop, $R_i$ is obtained by subtracting the final $T_i$ and $S_i$ from the raw data. Note
that the trend term obtained here is a locally fitted function, so it is completely different from the regression parameter obtained

by a liner regression, which will later be called the trend.

The STL method has five algorithmic parameters: the number of iterations in the inner loop, $n_i$, the number of iterations in
the outer loop, $n_o$, the smoothing parameter of the low-pass filter, $n_l$, the smoothing parameter of the trend component, $q$, and
the seasonal smoothing parameter, $n_s$. The value of parameter $n_l$ is imposed by the time series sampling frequency and set here
to 365, which is the least odd integer greater than or equal to the time series frequency. The parameters $n_i$ and $n_o$ are set to the

recommended values, i.e. $n_i = 1$ and $n_o = 15$ (Cleveland et al., 1990). Following the same recommendation, the parameter $q$
is defined as the first integer respecting the following condition:

$$q \geq \frac{1.5 n_p}{1 - 1.5 n_s^{-1}} \tag{4}$$

where $n_p$ is the time series frequency.

For the the seasonal smoothing parameter $n_s$, no formal recommendation based on previous mathematical analyses exists

(Cleveland et al., 1990). This parameter determines the variation of the seasonal signal over time and thus the fraction of the
data variation that is included in the seasonal component. If set to a small value, the seasonal component will highly vary from
year to year, including inter-annual variability. If set to a too large value, the seasonal component will be completely periodic
from year to year, and the method is no longer superior to a simple periodic removal of the seasonal signal (as classically done
in hydrological time series analysis, e.g. in the work of Schaefli et al. (2007)). The method proposed by Cleveland et al. (1990)

to adjust this parameter is not applicable here: it would require an assessment based on 365 different plots per catchment. We
propose here to use the auto-correlation function (ACF) and the partial auto-correlation (PACF) of the residuals time series to
select an appropriate $n_s$. In fact, the ACF and the PACF can be used to ensure that no seasonality remains in the residuals. The
ACF and the PACF of the residuals should in particular not show any significant correlation at the half-annual (183 days) or
the annual scale (365 days), since this would be indicative of seasonal components being left in the residuals.

Therefore, the following method is applied to all water temperature, discharge, air temperature and precipitation time series:
the STL is run for $n_s$ ranging from 7 to 99 (note that $n_s$ has to be odd and $\geq 7$), and the ACF and PACF are computed for
all residuals time series. The mean ACF and PACF values for lags between 360 and 370 are plotted against the $n_s$ value and
the plots are checked individually by visual inspection to determine the best $n_s$. Visual inspection is justified by the fact that





for some catchments and variables, the PACF decreases monotonically and tends to a constant value, whereas in other cases, it reaches a minimum before increasing again, making an automated decision process difficult. Based on this analysis, the value retained for this study is $n_s = 37$, for all variables and all catchments. A single value for all catchments and variables is preferable. Indeed, since this value defines how the signal can evolve over time, and thus influences the trend and the residuals

terms, different values would make the comparison of linear regression output between catchments and variables less relevant.

Some example output of the STL method and additional details are given in Section S1.3 of the supplementary. It is noteworthy that the de-seasonalization with the STL method has almost no effect on precipitation. However, in Figure S4 in supplementary, we show that the seasonality in precipitation time series is weak.

### 3.3   Linear regression

The temporal trends are explored using linear regressions over different time periods, which has been shown to be a suitable approach (Lepori et al., 2015) and is commonly used (Hari et al., 2006; Schmid and Köster, 2016). A linear regression is applied to all four de-seasonalized variables (i.e. $T_i + R_i$ from the STL method, for the variables water temperature, discharge, air temperature and precipitation) against time with the classical least squares estimation technique. The linear model is applied, when possible, for the periods 1979-2018 and 1999-2018.

As expected, the correlation determination $R^2$ values are relatively low, because the daily and inter-annual variability is still present in the time series and the linear model cannot represent these components. However, the p-values are all very small and the residuals of the linear model shows that, for all periods, the linear regression against time only is a suitable estimator of the time series evolution.

The linear regressions are also applied to seasonal and annual mean time series. In this case, the $R^2$ values are clearly

higher, since there is less variance in the input data, but the p-values increase. Indeed, only 20 or 40 points are used depending on the time period, reducing the robustness of the method. Some p-values are even above the significance threshold. As a consequence, the long-term trends presented in this paper only use de-seasonalized time series, with p-values<0.05. Seasonal trends, obtained from seasonal mean values, must be interpreted cautiously. In the seasonal case, most of the analysis is based on raw seasonal means instead of trends because of their low significance level.

For catchments with more than one meteorological station attributed, the trends used in the analysis for air temperature and precipitation are the mean of the trends of all the catchment's stations. For precipitation and discharge, they are expressed in relative changes to allow for a comparison between catchments. Unless stated explicitly, trends are expressed per decade.

### 3.4   Ecological indicators

Two important ecological indicators are used to quantify the impact of river warming and its evolution: the number of days

during which stream temperature reaches or exceeds the value of 25°C, and the number of consecutive days during which the hourly temperature remains above 15°C.

The 25°C threshold is a legal limit in Switzerland above which heat release in rivers is forbidden; this is important for example for nuclear power plant cooling. The indicator is computed as follows: based on hourly data, when the water temperature





reaches 25°C at least for one hour, the day is flagged as above 25°C. Then, the number of such days per year are summed in order to investigate the evolution over time.

The 15°C threshold is important for fish health. Indeed, the Proliferative Kidney Disease (PKD) affecting salmonid fish is caused by a parasite that proliferates when water temperature remains above 15°C for a few weeks (Hari et al., 2006; Carraro
et al., 2016, 2017). Water temperature affects the impact of PKD and its prevalence (Carraro et al., 2017).

The indicator is computed following a simple approach inspired by the more complex model proposed in Carraro et al. (2016). First, the days during which the water temperature remains above 15°C for the whole day are computed (a 4 hours moving window average is applied beforehand). Then, data are filtered to keep only series longer than 28 consecutive days. Finally, the number of days above 28 in the remaining series are summed for each year. The results indicate the number of days
in the year for which the temperature is above 15°C for at least 28 consecutive days. The process behind PKD being far more complex, this method does not pretend to be exact in determining the presence or absence of PKD in monitored rivers, but is an indicative approach to assess the exposure evolution of the river system. A sensitivity analysis has been performed and the qualitative evolution is not dependent on the chosen values.

## 4 Results and discussion

### 4.1 Long-term evolution of stream temperature and discharge

The water temperature evolution for all gauging stations used in the current study is shown in Figure 2 top panel. In spite of the high natural variability, a warming trend is visible in most rivers. To investigate this evolution in detail, catchments with temperature measurements available since 1970 have been selected (14 catchments). Figure 2 bottom panel shows the temperature anomalies per decade with respect to the 1970-2018 mean for these catchments. A two-sided t-test is performed to
assess if the differences in decadal means are significant. Except between the 1970's and 1980's, where no significant difference is found (p-value = 0.17), all other anomaly means are shown to be one-by-one significantly different (p-values $< 5 \cdot 10^{-5}$ for the three tests) which confirms the important rise observed since 1980 (Figure 2 bottom panel). The shift occurring at the end of the 1980's reported by Hari et al. (2006) and discussed in Figura et al. (2011) and Lepori et al. (2015) is not observed in all rivers (see Figure 2). Indeed, the shift is clearly visible in catchments located on the Plateau/Jura and downstream lakes, but not
necessarily in alpine catchments or catchments strongly influenced by hydropeaking. Note that this shift is also present in air temperature records (see Figure S9 in supplementary). The shift between the 1980's and 1990's decade is more important than previous or subsequent shifts, but contrary to the statement in Hari et al. (2006), the warming trend continues after the shift. Looking at the 30 years anomaly difference, the mean anomaly difference over the 14 catchments for the period 1970-2000 is of 0.59 °C and for the period 1990-2018 is of 0.55°C. A partially overlapping samples two-sided t-test (Derrick et al., 2017)
finds no significant difference between these two values (p-value = 0.59, this test is used instead of a classical t-test since the two samples overlap). Consequently, the "end-of-80's" shift might be interpreted as a hiatus in the long-term trend. The apparent acceleration of the warming seen over the last years is due to the extreme year 2018 which pulls up the running mean.

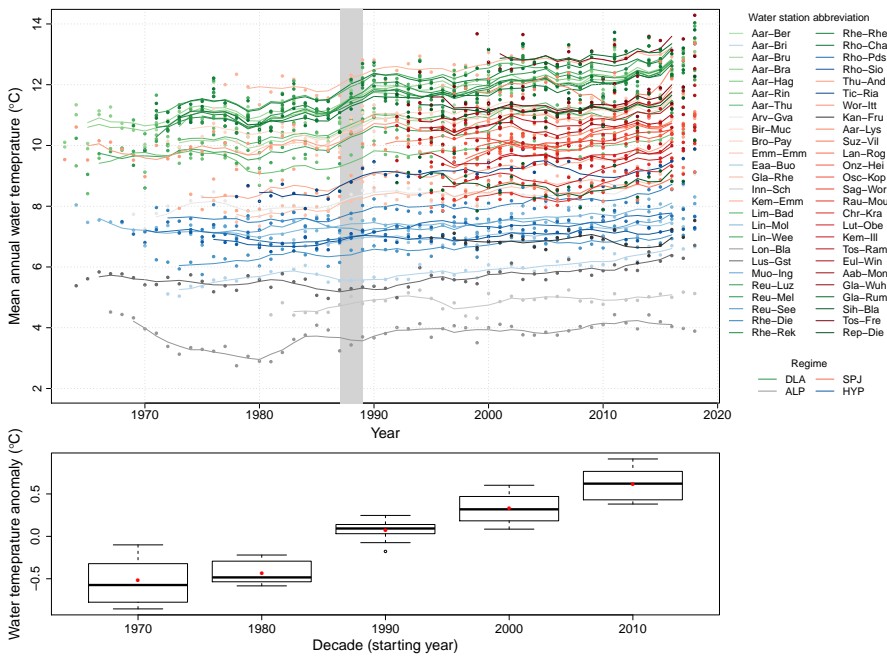

**Figure 2.** Top: Mean annual stream temperature of the 52 catchments described in Table 1. Lines show the 5-years moving averages. Colours indicate the hydrological regimes. The 1987/1988 transition period is highlighted in grey. Abbreviation for river names are given in Table 1 and abbreviation for regimes are: DLA = downstream lake regime, ALP = alpine regime, SPJ = Swiss Plateau/Jura regime and HYP = strong influence from hydropeaking. Bottom: Water temperature anomalies per decade with respect to the 1970-2018 mean, for the 14 catchments with data available since 1970. Thick lines are the median and red dots the mean values (values used for the t-test and the partially overlapping samples t-test, see text). Boxes represent the first and third quartiles of the data, whiskers extend to points up to 1.5 time the box range (i.e. up to 1.5 time the first to third quartiles distance) and extra outliers are represented as circles.

A long-term analysis is also performed on discharge data (Figure 3). In this case, catchments with measurements ranging back to at least 1920 (20 catchments) are kept for anomaly analysis. Figure 4 shows that there is almost no trend on the long-term for annual mean discharge and precipitation (for the discharge the mean trend obtained by linear regression over the 26 catchments available between 1970 and 2018 is of -0.5 % per decade). However, the recent decades show a clear negative trend.

5    The 1980's decade exhibits a positive runoff anomaly with a decrease toward the end of the decade. This discharge surplus at the beginning of the 80's has partially mitigated the warming early in the decade and the runoff decrease observed at the end of the decade has probably contributed to the temperature shift. However, shifts in runoff are also seen at other times (e.g beginning of the 70's), without a large impact on water temperature. A 7-8 years cycle in runoff annual mean can be seen in Figure 3. It is related to the cycle found in the North Atlantic Oscillation (NAO) and the Atlantic Multi-decadal Oscillation

10   (AMO) which has already been discussed in the literature (Lehre Seip et al., 2019). These time series are presented in Figure S10 in supplementary. This cycle seems to have no real impact on stream temperature as it is not visible in Figure 2.





A longer multi-decadal variation can be seen in discharge data (see Figure 4). However, one century of data is not long enough to assess if there is a real 30-40 years cycle, which could be related to the 34-36 year cycle found in the Atlantic Meridional Overturning Circulation (AMOC) (Lehre Seip et al., 2019), or if there is only some statistical variation. As a consequence, it is not possible to assess if the decrease over the last decade is part of a long-term cycle or results from climate change, or both.

The decades 1970-1980 and 1980-1990 show a more marked anomaly (negative first and then positive, see Figure 4) for discharge than for precipitation. This is explained by the glacier melt evolution, which reaches a minimum in the 1970-1980 decade followed by a sharp increase in the next decade (Huss et al., 2009).

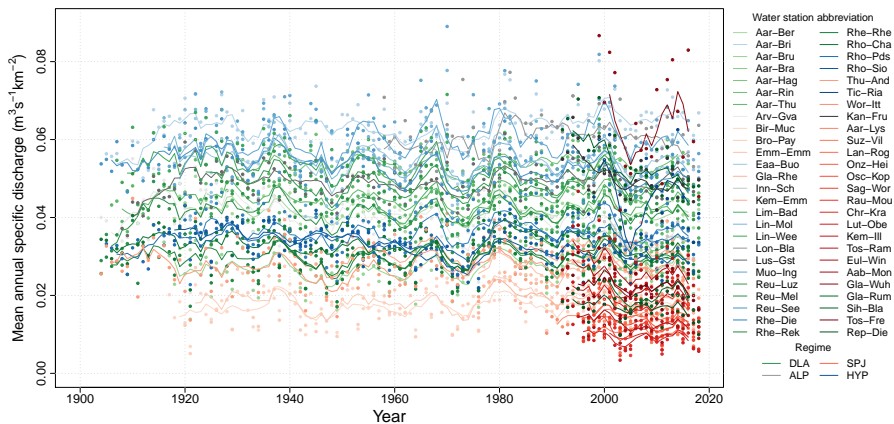

**Figure 3.** Mean annual specific discharge for the 52 catchments described in Table 1 (normalized by catchment area for comparison). Lines show the 5-years moving averages. Colours indicate the hydrological regimes.

## 4.2 Temperature and discharge trends from linear regression

The trends in stream temperature and discharge have been computed with linear regression over the period 1999-2018 for all 52 catchments and over the period 1979-2018 when possible. All trend values are presented in the Appendix in Tables A1 and A2 for water temperature and discharge, and in Tables S3 and S4 in supplementary for air temperature and precipitation. The plots shown in this section are for the period 1999-2018, where more catchments are available. Similar plots for the period 1979-2018 are shown in Figures S11 and S12 in supplementary. Note that results presented in this section, except for the trends in runoff in the last decades, also hold for the longer time period, and the results are even more evident on this longer time period. This can be explained by the lower sensitivity to boundary conditions and overall highest robustness of linear regressions over longer time periods.

Trends in stream temperature and discharge are compared to trends in air temperature and precipitation in Figure 5. There is a clear increase in water temperature and a reduction in discharge observed in Swiss rivers over the 1999-2018 period. The mean trends for the last 20 years are +0.37°C per decade for water temperature (with a large spread in the distribution), +0.39°C





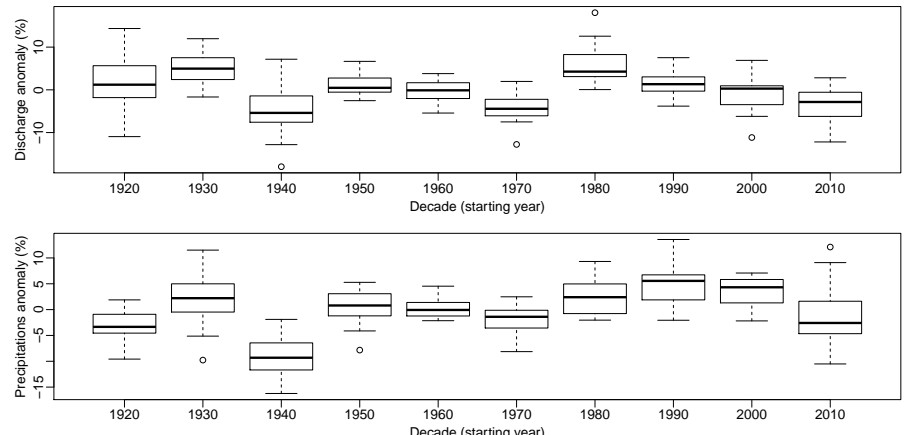

**Figure 4.** Relative discharge (top) and precipitation (bottom) decadal means of anomalies with respect to the 1920-2018 average for 20 catchments and 17 MeteoSwiss homogeneous stations with data available since 1920 (see Table 1 and Table S2 in supplementary).

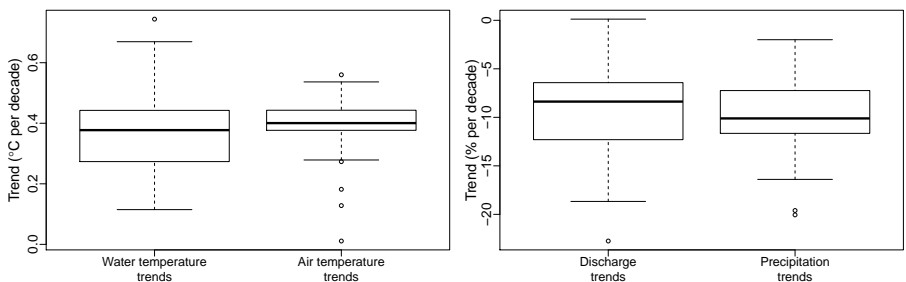

**Figure 5.** Water and air temperature trends (left), normalized discharge and normalized precipitation trends (right), for the period 1999-2018 and for the 52 catchments described in Table 1 and their associated meteorological stations.

per decade for air temperature, -10.1% per decade for discharge and -9.3% per decade for precipitation. However, the trends in precipitation and runoff have to be considered with caution regarding the long-term variation discussed above. For the period 1979-2018, the trends are the following: +0.33°C per decade for water temperature (with a large spread in the distribution), +0.46°C per decade for air temperature, -3.0% per decade for discharge and -1.3% per decade for precipitation.

5    The water temperature and discharge trends for the four different regimes are shown in Figure 6. A two-sided Wilcoxon test is used to assess whether differences between regimes are significant in terms of temperature trends (results shown in Table 2). Since some categories have only a few observations and normal distribution can not be assumed, this test is used instead of a t-test. Two groups can clearly be identified: downstream of lakes (DLA) and Swiss Plateau-Jura (SPJ) regimes on the one hand, and alpine (ALP) and hydropeaking influenced (HYP) regimes on the other hand. Indeed, for both pairs, the



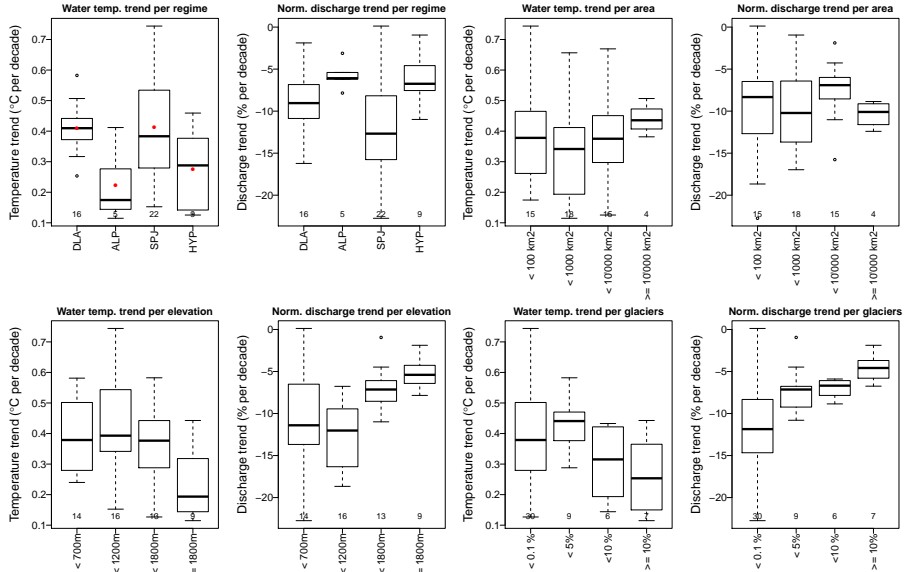

**Figure 6.** Water temperature and discharge trends for the period 1999-2018. Top left: classified upon the four different hydrological regimes (DLA = downstream lake regime, ALP = alpine regime, SPJ = Swiss Plateau/Jura regime and HYP = strong influence from hydropeaking). Top right: classified upon the catchment area. Bottom left: classified upon the catchment mean elevation. Bottom right: classified upon the glacier coverage. The numbers at the bottom indicate the number of catchments in each category. On the top left boxplot, red dots are the mean values (values used for the Wilcoxon test, see text).

hypothesis of different mean is clearly rejected with p-values>0.15. The water temperature trends are significantly lower for alpine catchments and catchments strongly influenced by hydropeaking. The impact of lakes is discussed in Section 4.3.

**Table 2.** P-values of Wilcoxon two-sided test between the trends in water temperature for the four hydrological regimes, period 1999-2018 (left) and 1979-2018 (right).

| | Period 1999-2018 52 catchments | | | | Period 1979-2018 28 catchments | | |
|---|---|---|---|---|---|---|---|
| | **ALP** | **SPJ** | **HYP** | | **ALP** | **SPJ** | **HYP** |
| **DLA** | 0.008 | 0.672 | 0.031 | **DLA** | 0.005 | 0.18 | 6.3e-6 |
| **ALP** | - | 0.019 | 0.519 | **ALP** | - | 0.024 | 0.833 |
| **SPJ** | - | - | 0.046 | **SPJ** | - | - | 0.002 |

The catchment area has no clear influence on trends (see Figure 6) despite that area is clearly correlated with the regime (Table 1). To infer the isolated effect of area, only catchments from Plateau/Jura regimes are used (largest sample of rivers, no major disturbance), but no correlation between water temperature or discharge trends and area can be found (see Figure S13 in supplementary).





Elevation and the fraction of glacier coverage in the catchments (which are strongly correlated) clearly influence water temperature and discharge trends (see Figure 6 lower panels). Lowland catchments, mostly located in the Plateau and Jura regions, experience the most important decrease in discharge. At higher elevation, the loss of ice mass from glaciers appears to be the most important factor counteracting decrease in discharge during the last 20 years. Snow and glacier melt also tend to

moderate the warming of streams in high-elevation catchments. For these reasons, discharge and temperature of alpine streams are the least impacted by climate change until now. However, if this buffer effect induced by glaciers and seasonal snow cover disappears due to continuation of temperature rising in the future (Bavay et al., 2013; Huss et al., 2014; MeteoSuisse et al., 2018), the alpine catchments will be amply impacted (see Section 4.4.4).

Unsurprisingly, rivers strongly influenced by hydropeaking show lower trends compared to undisturbed ones. This results

from large volumes of cold water being released from reservoirs located at high elevation to lowland rivers as discussed for instance in Feng et al. (2018).

In conclusion, for Swiss Plateau and Jura catchments, air temperature seems to be the main driver, and the mean of the trends for this type of catchment is close to the mean air temperature trend. For alpine catchments and catchment strongly influenced by hydropeaking, additional factors, such as snow and glacier melt and anthropogenic disturbances gain importance in the

energy balance, superseding the dominance of the air temperature as driver for the water temperature.

### 4.3 Effect of lakes

In the previous section, it was shown that rivers located downstream of lakes have water temperature trends similar to Swiss Plateau and Jura catchments, in spite of an higher mean elevation and a larger glacier-covered fraction (see Table 1), which typically attenuate the water temperature increase.

The effect of lakes located at the foot of mountain ranges on stream temperature is well known. The input water originates from alpine rivers (potentially disturbed by hydropeaking), which are colder than the surrounding environment and not in equilibrium with local air temperature. Since water has a certain residence time in the lake, its temperature increases due to atmospheric forcing and the main driver for outflow water temperature is the air temperature (see Figure 3). However, it has not demonstrated yet if the effect of lakes on river temperature trends is similar. In Schmid and Köster (2016), it is shown that

due to solar brightening lake temperature trends can exceed air temperature trends.

To investigate the effect of lakes on water temperature trends, five lake systems with measurements at the inflow and at the outlet are selected: Thun-Brienz lakes system, Lake Biel, Lake Luzern, Lake Walen and Lake Geneva. Temperature anomalies with respect to the period 1979-2018 and trends are plotted for water temperature at each station and air temperature at meteorological stations representative of the catchment. The results are shown in Figure 7 for Lake Geneva and in Figures S14

to S17 in supplementary for the other four lakes. The trends for the different inflow and outflow rivers and for air temperature are presented in Table 3.

For Lake Walen and Lake Geneva, the effect is obvious: the outlet trend is almost equal to the collocated air temperature trend. Even if trends on inflows are much smaller, they do not significantly influence the outlet waters (see Table 3). The lake acts as catalyst and the system reaches a quasi-equilibrium. For Lake Geneva, the water temperature of the Arve river is also





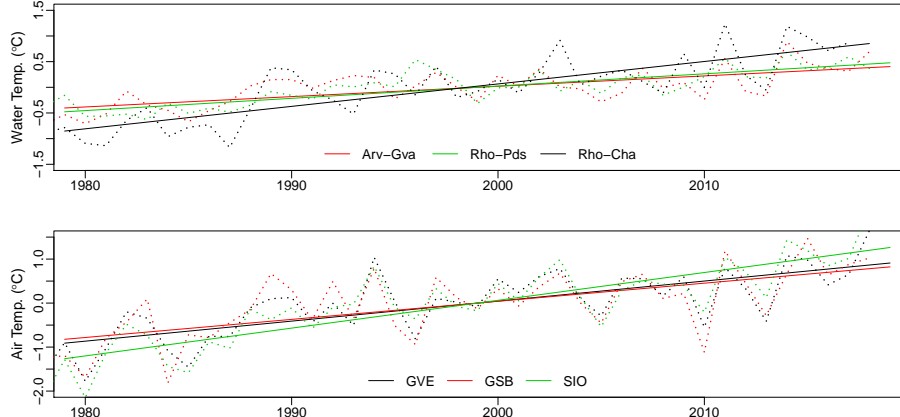

**Figure 7.** Top: Lake Geneva, Water temperature anomalies and trends for inflow (Rho-Pds) and outlet (Rho-Cha) stations (top). Arv-GVA denotes the Arve in Geneva, Rho-Pds the Rhone in Porte du Scex, and Rho-Cha the Rhone in Chancy. Bottom: Air temperature anomalies and trends for surrounding MeteoSwiss stations (bottom). GVE denotes Geneva-Cointrin, GSB Grand Saint-Bernard and SIO Sion. The period for trend computation is 1979-2018.

**Table 3.** Inflow and outflow water temperature trends for 6 different lakes, air temperature trends for stations in or close to the lake catchments, and trends for additional catchments mentioned in the text. Period for trend computation is 1979-2018, except for the Engelberger Aa in Buochs where the trend is computed over the period 1999-2018 because of limited data availability.

| Lake | Inflow station | Inflow trend (°C per decade) | Outflow station | Outflow trend (°C per decade) | Meteo stations | Air temp. (°C per decade) | Additional stations | Add. stat. trend (°C per decade) |
|---|---|---|---|---|---|---|---|---|
| Geneva | Rhone in Porte du Scex | 0.24 ± 0.01 | Rhone in Chancy | 0.44 ±0.01 | GVE | 0.46 ± 0.02 | Arve in Geneva | 0.20 ± 0.01 |
| | | | | | GSB | 0.41 ± 0.03 | | |
| | | | | | SIO | 0.63 ± 0.02 | | |
| Wahlen | Linth in Mollis | 0.24 ± 0.01 | Linth in Weesen | 0.44 ±0.01 | GLA | 0.44 ± 0.03 | - | - |
| | | | | | ELM | 0.48 ± 0.03 | | |
| | | | | | SMA | 0.46 ± 0.03 | | |
| Luzern | Muota in Ingenbohl | 0.08 ± 0.01 | Reuss in Luzern | 0.48 ± 0.01 | ALT | 0.48 ± 0.02 | Kleine-Emme in Emmen | 0.42 ± 0.01 |
| | Reuss in Seedorf | 0.19 ± 0.01 | | | ENG | 0.43 ± 0.03 | | |
| | Engelberger Aa in Buochs | 0.29 ± 0.02 | | | LUZ | 0.48 ± 0.02 | | |
| Brienz | Aare in Brienzwiler | 0.24 ± 0.01 | Aare in Ringgenberg | 0.31 ± 0.01 | MER | 0.50 ± 0.02 | - | - |
| | | | | | GRH | 0.43 ± 0.03 | | |
| | | | | | INT | 0.52 ± 0.02 | | |
| Thun | Aare in Ringgenberg | 0.31 ± 0.01 | Aare in Thun | 0.37 ± 0.01 | MER | 0.50 ± 0.02 | - | - |
| | | | | | BER | 0.48 ± 0.02 | | |
| | | | | | INT | 0.52 ± 0.02 | | |
| Biel | Aare in Hagneck | 0.49 ± 0.01 | Aare in Brugg | 0.43 ± 0.01 | BER | 0.48 ± 0.02 | - | - |
| | | | | | CDF | 0.49 ± 0.03 | | |
| | | | | | WIN | 0.44 ± 0.02 | | |





shown. The Arve river originates from the Mont-Blanc massif (France) and flows for about 100 km through the Arve valley before joining the Rhone in Geneva. Despite flowing through low-lying land, the Arve keeps its alpine characteristics whereas these characteristics are completely lost in the Rhone river after the lake.

In Lake Luzern, a similar effect is observed. Indeed, the three rivers feeding into the lake (Reuss, Muota, and Engelberger

Aa) show trends which are considerably lower than for the Reuss river in Luzern (see Table 3). However, the Kleine-Emme, which joins the Reuss just after Luzern, shows a similar trend without any lake present along its course, demonstrating that, for a mid-elevation stream, flowing a certain distance in the Plateau leads to similar effect as induced by lakes. For the Lakes Thun-Brienz system, the water temperature trend is enhanced as a result of the two subsequent lakes and it tends to the air temperature one.

For Lake Biel, no effect is observed. This is not surprising since the Aare input water has already a trend similar to the local air temperature trend. In addition, the residence time in Lake Biel is very short (58 days, while for the five other lakes it ranges from 520 to 4160 days (Bouffard, 2019)), limiting the exposure time of lake waters to atmospheric forcing. This has been described in more details in Råman Vinnå et al. (2017).

In conclusion, despite their higher mean catchment elevation, water temperature trends for stations at lake outlets are similar

to Plateau trends. Lakes having much longer residence times for water than rivers, they are smoothing out local effects such as snow or glacier melt or precipitation and have air temperature as main temperature forcing. As a consequence, water temperature trends at the outlet of lakes are therefore, in general, similar to air temperature trends (and to trends observed on the Swiss Plateau).





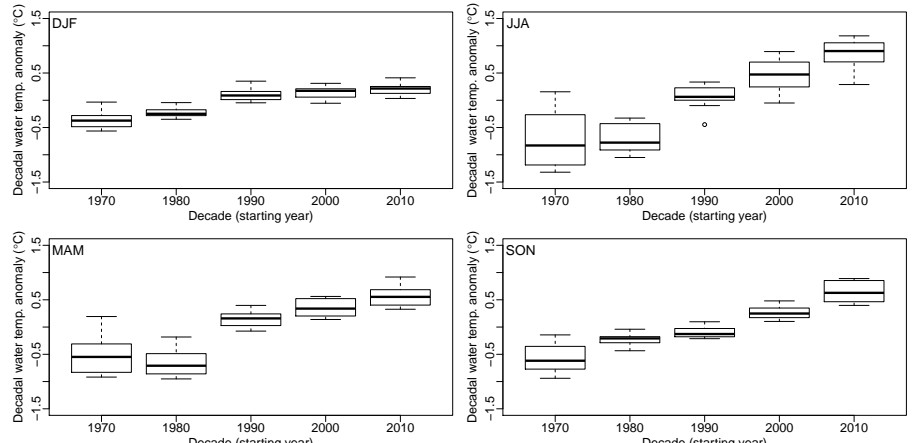

**Figure 8.** Water temperature seasonal anomalies for the 14 catchments where data are available since 1970 (see Table 1). Anomalies with respect to the 1970-2018 period.

## 4.4 Seasonal trends and relation with air temperature and precipitation

In this section, stream temperature and discharge trends and anomalies are analysed at seasonal scale. The relation between these two variables and the meteorological conditions (air temperature and precipitation) are also discussed on a seasonal basis. Then, particular seasonal features are addressed. Finally, the evolution of the infra-annual variability along with the

inter-seasonal correlation, or system memory, are discussed. Even if the inter-variable correlation and system memory are not directly linked to observed changes, they are key factors to understand the system dynamics and thus, are essential to infer impacts of climate change on water temperature and discharge. The analysis below is mostly based on the 1999-2018 period. Seasons are defined as follows: winter is December-January-February (DJF), spring is March-April-May (MAM), summer is June-July-August (JJA) and fall is September-October-November (SON).

long-term evolution of the seasonal anomalies are shown in Figures 8 and 9 for water temperature (decades 1970 to 2010) and discharge (decades 1960 to 2010). Air temperature and precipitation are shown in Figures S18 and S19 in supplementary and exhibit similar behaviour. For all seasons, the water temperature is significantly rising since 1980. The warming is more important in summer and less pronounced in winter. For discharge, spring and fall do not show an obvious trend on the long-term. There is a clear decrease in summer since 1980 while winter shows a slight increase.

Annual and seasonal trends for stream and air temperature, discharge and precipitation are presented in Figure 10 for the period 1999-2018. They confirm the tendencies described above. Mean water temperature trends are slightly smaller than air temperature trends for all seasons except for spring when they are notably larger. This shows that rivers do not react linearly to a general warming of the atmosphere and additional factors are controlling these complex systems. For discharge, negative trends are found in all seasons except for winter when they are almost null. Discharge trends follow precipitation trends in all

seasons. In general, precipitation determines the discharge trend and consequently, snow and glacier melt play a minor role in





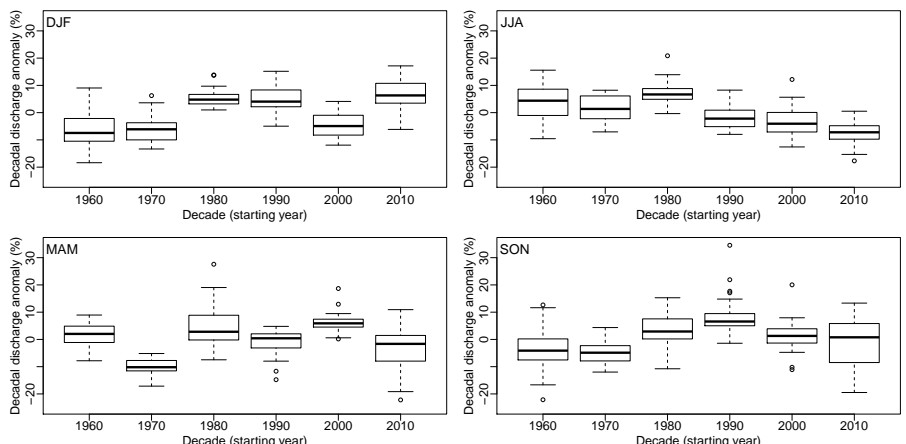

**Figure 9.** Discharge seasonal relative anomalies for the 26 catchments where data are available since 1960 (see Table 1). Anomalies with respect to 1960-2018 period.

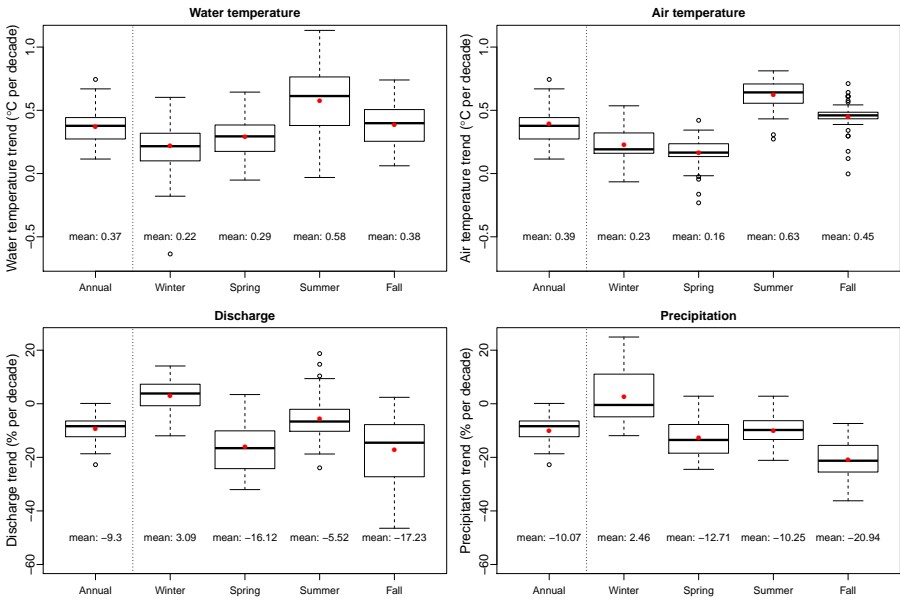

**Figure 10.** Annual and seasonal trends for water temperature, air temperature, discharge and precipitation. The mean values are indicated by red dots and written below boxes in °C per decade or % per decade.





the observed trends. However, for specific catchments, this can be different. When looking at individual catchments, there is only a insignificant correlation between trends in air and water temperature, and between trends in discharge and precipitation (see Table S5 in supplementary). This absence of correlation results from the noise in the individual trend values due to the short time period available. This is a limitation of the method applied and thus trends can not be used for an inter-variable interaction study.

To explore the correlation between variables, raw values are used. Table 4 shows the correlation between main variables on a yearly and seasonal basis. These values are obtained by computing correlation of two variables for individual catchments and then averaging these correlations. As a measure of the robustness of the method, the number of catchments where correlation is insignificant (p-value > 0.05) is indicated. At annual scale, air temperature is the main driver of water temperature. The negative correlation between water temperature and discharge is rather weak and not significant in almost half of the catchments. As expected, discharge and precipitation are strongly correlated.

**Table 4.** Correlation between the annual and seasonal time series of water and air temperature (left), water temperature and discharge (middle) and discharge and precipitation (right). Correlations are computed for all 52 individual catchments over the period 1999-2018 and then averaged over all catchments. Numbers in parenthesis indicate the number of catchments where the correlation is not significant (p-value>0.05 for the null hypothesis being no correlation).

| Water and air temperature | | Water temperature and discharge | | Discharge and precipitation | |
|---|---|---|---|---|---|
| Period | Cor. | Period | Cor. | Period | Cor. |
| Annual | 0.77 (3) | Annual | -0.44 (24) | Annual | 0.73 (6) |
| Winter | 0.73 (1) | Winter | 0.27 (37) | Winter | 0.64 (9) |
| Spring | 0.76 (2) | Spring | -0.51 (19) | Spring | 0.66 (12) |
| Summer | 0.61 (7) | Summer | -0.66 (9) | Summer | 0.55 (10) |
| Fall | 0.76 (3) | Fall | -0.20 (40) | Fall | 0.64 (8) |

### 4.4.1 Winter

The water temperature trends in winter are the lowest of the four seasons and the discharge exhibits a slight positive trend, opposed to the negative discharge trend in all other seasons (see Figure 10). The positive trend in winter discharge is mainly

driven by the increase in winter precipitation. This is the season where the precipitation and discharge trends are the closest and the correlation between precipitation and discharge is strong and significant (see Table 4).

There is a weak positive correlation between winter discharge and winter water temperature. Even though this correlation is not significant in the majority of the catchments, it indicates a different behaviour compared to spring and summer. An explanation could be that increased water input during winter causes a push of relatively warm groundwater. Catchments with

increased winter discharge would thus have a more pronounced temperature trend. In contrast, some catchments show negative water temperature and discharge trends in winter (see Appendix Table A1). In this case, the lower discharge favours a more





pronounced water cooling through heat exchange and this effect might compensate and even overcome the air temperature trend. Both of these effects would lead to a positive correlation. The annual anomalies in winter water and air temperature, discharge, and precipitation are presented in Figure S20 in supplementary.

### 4.4.2 Spring

In spring water temperature trends are more pronounced than air temperature trends (Figure 10). Looking at individual catchments indicates that the most affected ones are mainly low-lying, non-glacierized SPJ catchments (see Appendix Table A1). These catchments experience the most significant discharge decrease in spring, probably due to an earlier snow melt period, which possibly explains their higher sensitivity to air temperature. Indeed, snow melt releases cold water acting as a buffer and reducing the sensitivity to air temperature. Figure 11 shows the yearly anomalies in spring. The air temperature remains the main driver, however high discharge (e.g. 1999 or 2006) or low discharge (e.g. comparing 2013 and 2015) conditions have a clear anti-correlated impact on water temperature too. This can be seen in the negative correlation between air temperature and discharge in spring (Table 4).

A likely impact of climate change is an earlier and shorter snow melt season. Figure 12 shows the evolution of snow melt in terms of snow water equivalent (SWE) in spring over the last 20 years for Switzerland. There is no clear long-term trend in the total spring melt and therefore, no contribution to the discharge trend. However, snow melt remains a key factor for spring discharge. For example, in 1999, 2009, 2012 and 2018, precipitation deficits are well compensated by the above-average snow melt, while in 2002 and 2007, the opposite effect is observed. Such discharge variations have a direct impact on water temperature.

### 4.4.3 Summer, extremes, and fall

Summer exhibits the strongest positive water temperature trends and negative discharge trends, both on the past 20 and 40 years (see Figure 10 and Appendix Tables A1 and A2). It also has the weakest correlation between water and air temperature and the strongest negative correlation between water temperature and discharge (see Table 4), indicating that summer is the season when water temperature is the most sensitive to discharge. Also, correlation between precipitation and runoff is lowest in summer. This is likely due to the role of evapotranspiration in summer and the variability of the remaining snow at the beginning of summer (see Figure S21 in supplementary). There is a strong link between extremes in summer air temperature (2003, 2015, and 2018) and extreme summer stream temperature (see Figure 13), coinciding with a deficit in precipitation and in discharge. A positive air temperature anomaly in summer is generally associated with dry conditions in Switzerland (Fischer et al., 2007b, a). Sometimes, a below-average air temperature but an above-average water temperature is observed, e.g. in summer 2011. This is attributed to the lack of precipitation and the resulting runoff deficit. So, while precipitation deficit favors and enhances summer heat waves, it also has a direct impact on summer stream temperature. Another particularity is seen in 2013 and 2016, with a negative water temperature anomaly while the air temperature is close to the mean, likely induced by the above-average precipitation and runoff for these years. Therefore, the water temperature to discharge and precipitation negative correlation holds for both high and low values.





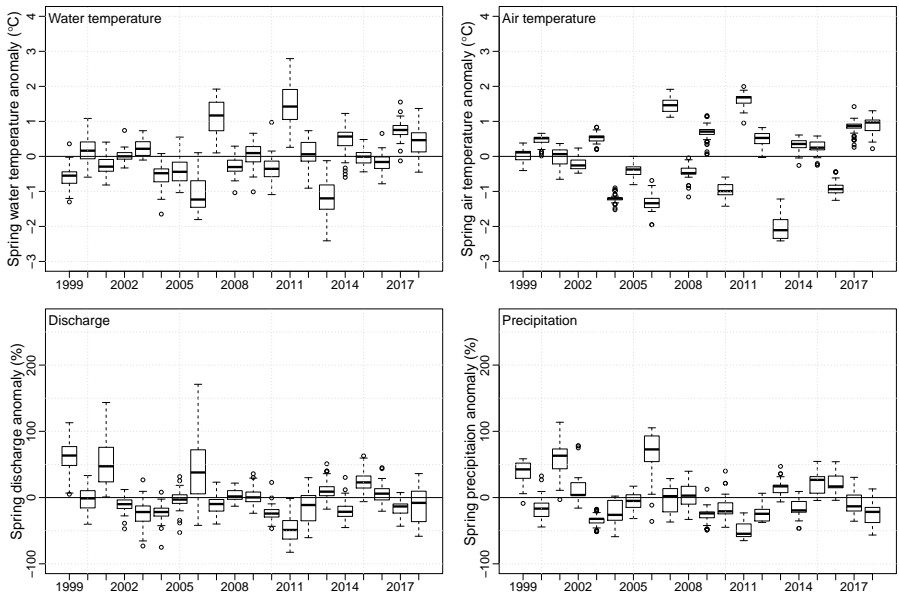

**Figure 11.** Spring anomalies in water temperature, air temperature, relative discharge and relative precipitation for all 52 catchments. Anomalies are computed with respect to the 1999-2018 mean for each catchment.

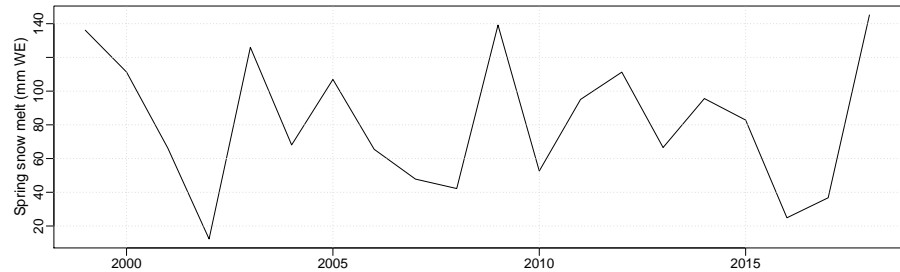

**Figure 12.** Snow melt evolution in spring in Switzerland, obtained by subtracting first of June SWE to first of March SWE. SWE for individual months from March to July is shown in Figure S21 in supplementary.





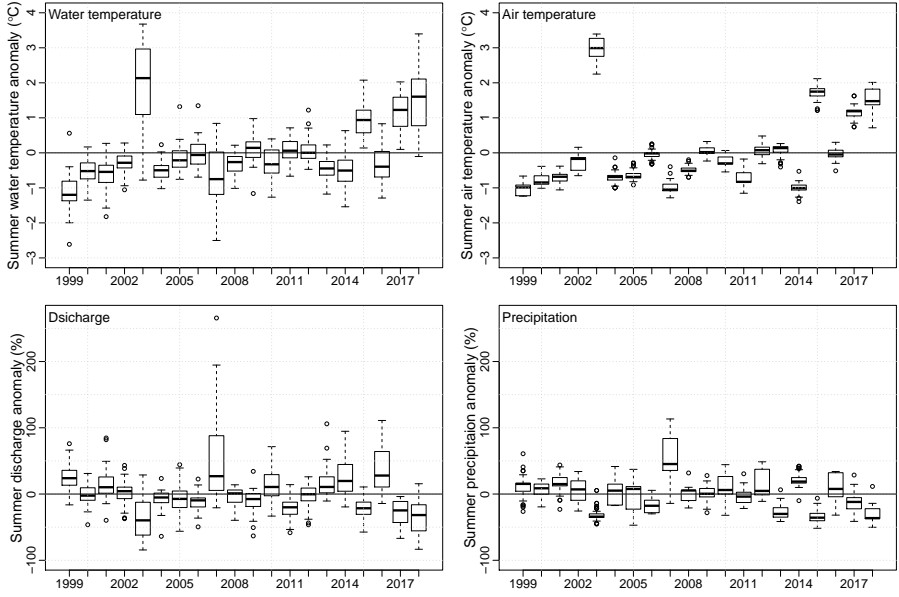

**Figure 13.** Summer anomalies in water temperature, air temperature, relative discharge and relative precipitation for all 52 catchments. Anomalies are computed with respect to the 1999-2018 mean for each catchment.

Summer snow melt, approximated by the amount of snow remaining at the beginning of June, shown in Figure S21, has an impact on summer stream conditions. Indeed, for high summer snow melt, (e.g. 1999, 2013) the runoff anomaly is positive and stronger than the precipitation anomaly. The opposite effect is seen in 2005 or 2011: the snow melt is low in summer, with a direct impact on stream temperature.

The anomalies in fall are presented in Figure S22 in supplementary. Discharge has a very low impact during this season. Since air temperature is the main driver, the inter-annual variability in fall is lower for water temperature than for air temperature.

### 4.4.4 The case of alpine catchments

The analysis in the previous sections has not considered the hydrological regime. However, alpine catchments show a particular behaviour. Over the last two decades, higher elevation catchments exhibit lower stream temperature trends along with less pronounced discharge decreases (see Figure 6) than lowland rivers. In winter, the air temperature trend is higher in the mountains than for the rest of the country, while the water temperature trend is smaller, showing the impact of enhanced snow melt induced by higher air temperatures, and thus cold water advection in rivers as discussed in Section 4.4.1. The same effect is seen in spring. In summer, the temperature trend is mainly driven by the local air temperature trend, which is lower than the median of the whole country, leading to a lower warming in alpine rivers than in lowland ones (see Figure 14 top part).

Alpine catchments are more preserved from extreme summer temperatures than other catchments (see years 2003, 2015, 2017 and 2018 in Figure 14 bottom part). Despite an important positive anomaly in air temperature, the water temperature





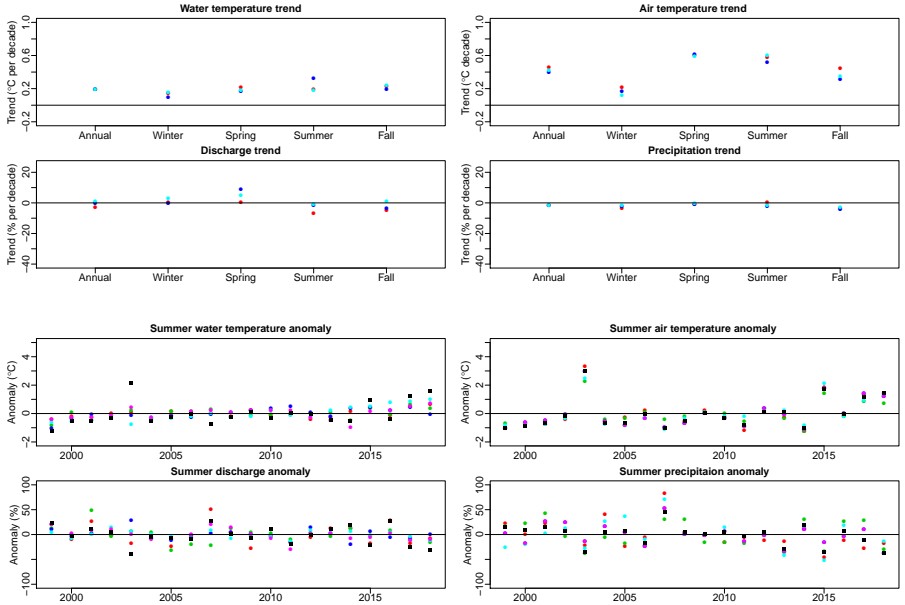

**Figure 14.** Top: Annual and seasonal trends for water and air temperature and for discharge and precipitation over the 1999-2018 period. Trends for the five alpine catchments (colour dots, denoted as Arve in Geneva (Arv-Gva), Inn in S-Chanf (Inn-Sch), Lonza in Blatten (Lon-Bla), Lütschine in Gsteig (Lut-Gst) and Kander in Frutigen (Kan-Fru)) and median for all 52 catchments (black square). Bottom: Summer anomalies for the same four variables, five catchments and period as on top. Median of the 52 catchments is also shown by a black square.

anomaly is considerably lower and below the median of catchments of other regimes. This resilience is attributed to many factors impacting alpine river temperatures such as geology, topography or permafrost (Küry et al., 2017) and, in the case of the extreme 2003 heat wave, by additional cold water released from glacier and snow melt during summer (Piccolroaz et al., 2018). This is confirmed by the positive or weak negative runoff anomaly over this year for alpine catchments whereas the

Swiss median anomaly in discharge is negative and the precipitation anomaly is clearly negative too (see Figure 14 bottom part). In addition, the peak in glacier melting is visible in glacier mass balance records (see Figure S23 in supplementary).

While this low sensitivity is obvious for 2003, when alpine catchments were almost not affected, the sensitivity seems more pronounced in 2015, 2017 and 2018. For these three years, the water contribution from glacier melt is lower, as shown by the mass balance of the GLAMOS glacier record (see Figure S23) and by the fact that discharge anomalies for these years are

closer to the mean of all catchments. Some catchments, e.g. the Lütschine in Gsteig, indicate that the way alpine streams react to summer air temperature and heat waves seems to change. This change is most probably induced by climate change. Note however, that the way alpine rivers responds to heat waves is a recent and not fully explored topic (Piccolroaz et al., 2018).

On the long-term, a shift of the thermal and hydrological regimes of alpine catchments is evident. As an example, Figure 15, obtained by averaging each day of the year (DOY) over an entire decade, shows a clear flattening of the discharge curve over

the last 50 years for the Lonza river (Glacier surface: 24.7%). Instead of a peak in the second half of the summer, the last two decades show a flatter discharge with a maximum at the end of June. In addition, the entire discharge distribution is shifted

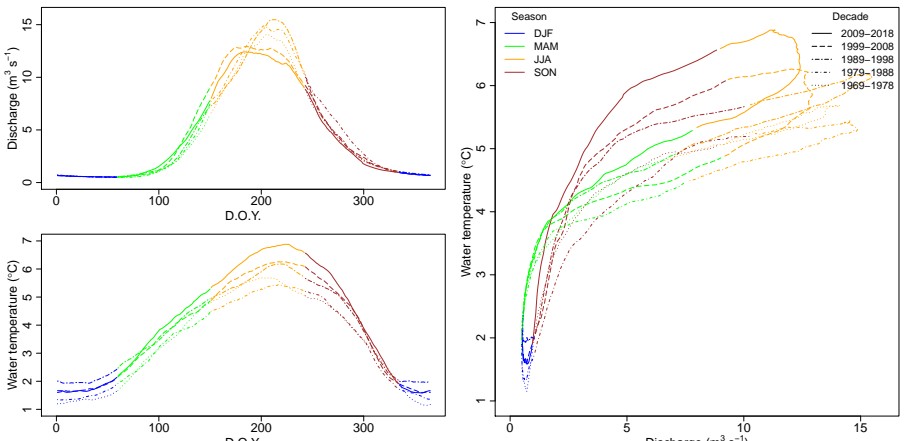

**Figure 15.** Left: Hydrological (top) and thermal (bottom) regimes per decade for the Lonza river at Blatten averaged for each day of the year (DOY). Line types represent decades and colours the seasons. Right: Decadal temperature plotted against decadal discharge (both averaged for each day of the year).

towards the beginning of the year, leading to an increase in spring and a decrease in late summer and autumn. There is a clear increase in water temperature, especially between mid-spring and mid-fall, which is stronger in the middle of the summer, leading to a wider temperature range spanned throughout the summer. This shift in hydrological regime and general warming significantly changes the evolution of water temperature versus discharge hysteresis curve. While in the 70's, the amplitude of hysteresis was rather limited (i.e. low sensitivity to summer air temperature), it becomes much wider during the last decades as a result of lower peak discharge and a higher water temperature. This is an additional evidence that alpine rivers are becoming more sensitive to climate change, potentially reacting in a strongly non-linear way in the future.

### 4.4.5 Infra-annual water temperature variability

With the summer water temperature trend being stronger than the winter trend, the infra-annual variability, i.e. the summer to winter temperature difference, is expected to increase over time. This will surely impact ecosystems, which will have to cope with warmer conditions and with increased variability. The topic of variability of air temperature under climate change is still an open discussion (Vincze et al., 2017). Figure 16, shows the annual difference between summer and winter means for all catchments with data since at least 1980. A 5-year moving average window is applied for noise reduction. Note the ∼14 years cycle present in the data with an amplitude of about 0.5 °C, probably caused by large scale atmospheric phenomena. The year-to-year variations of the temperature difference anomaly are more driven by this oscillation than by the underlying trend.

There is a clear evolution on the infra-annual variability: the computed trend indicates an increase of 0.3±0.1 °C per decade, which corresponds to a change of +1.2 °C over the studied period. The mean raw infra-annual variability equals 9.8 °C, with a standard deviation of 3.6 °C. This represents an increase of 10% to 20% of the variability for individual catchments. The





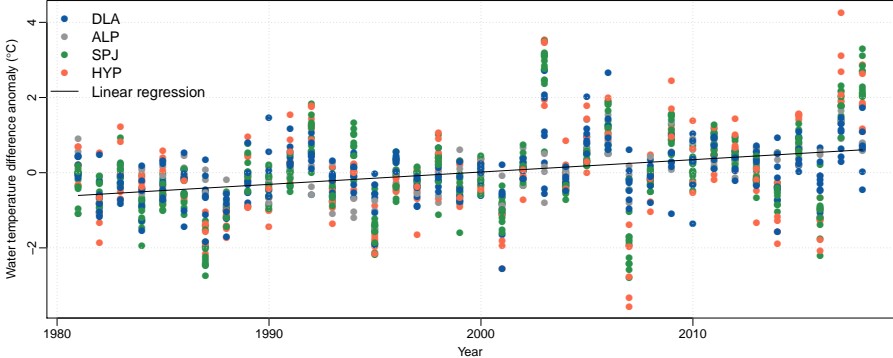

**Figure 16.** Water temperature summer to winter difference yearly anomalies. The hydrological regimes are represented by the dot colours and the abbreviations are Swiss Plateau and Jura regime (SPJ), Alpine regime (ALP), Downstream lake regime (DLA), and Regime strongly influenced by hydropeaking (HYP). The black line represents the fitted linear regression.

evolution of the summer to winter difference induced by the different seasonal warming rates is thus clearly important and must be considered for assessing the impact of climate change on ecosystems.

### 4.4.6 Inter-seasonal correlation and system memory

It is well known that the 2003 summer heat wave was enhanced by a long dry spell due to a precipitation deficit in late spring
5 and early summer (Fischer et al., 2007b). In this section, we explore if such robust seasonal connections exist with stream temperature and discharge. The seasonal relation can be studied by comparing Figures 11 and 13, and Figures S20 and S22 in supplementary. In addition, the correlations between water temperature and water temperature from previous seasons, between discharge and precipitation from previous season, and between water temperature and precipitation from previous seasons are shown in Table 5 and were obtained with the same method as the data in Table 4.

**Table 5.** Correlation for different seasons between water temperature and itself (left), precipitation and discharge (middle) and precipitation and water temperature (right). The correlations are computed between the season indicated in the line and the next season indicated in the column, (e.g. DJF and MAM show the correlation between winter and the next spring, while MAM and DFJ shows the correlation between spring and the next winter. Correlation between the same season shows correlation between seasons at one-year lag (i.e. correlation between winters and the next winters). The numbers in brackets indicate the number of catchments where the correlation is not significant (p-value>0.05 for the null hypothesis being no correlation).

| | Water temperature to water temperature | | | | | Precipitation to discharge | | | | | Precipitation to to water temperature | | | |
|---|---|---|---|---|---|---|---|---|---|---|---|---|---|---|
| | DJF | MAM | JJA | SON | | DJF | MAM | JJA | SON | | DJF | MAM | JJA | SON |
| DJF | 0.13 (50) | 0.34 (40) | 0.02 (50) | -0.09 (52) | DJF | -0.06 (52) | 0.18 (43) | 0.25 (45) | 0.09 (52) | DJF | -0.04 (51) | -0.15 (48) | -0.26 (48) | 0.05 (52) |
| MAM | 0.13 (52) | -0.18 (48) | 0.36 (36) | 0.04 (50) | MAM | -0.07 (48) | 0.05 (52) | 0.33 (32) | 0.27 (44) | MAM | 0.24 (48) | 0.36 (37) | -0.31 (36) | -0.11 (52) |
| JJA | 0.14 (50) | -0.08 (48) | 0.10 (48) | 0.28 (39) | JJA | -0.06 (50) | 0.19 (50) | -0.18 (51) | 0.31 (40) | JJA | -0.16 (50) | -0.06 (51) | 0.13 (47) | -0.22 (47) |
| SON | 0.28 (44) | 0.10 (47) | 0.09 (46) | -0.01 (50) | SON | 0.29 (44) | -0.02 (50) | 0.06 (51) | 0.05 (52) | SON | 0.14 (50) | 0.04 (52) | -0.07 (51) | -0.06 (52) |





Table 5 show that for water temperature there is almost no correlation and calculated values are mostly not significant. The only observed signal is from one season directly to the next one, but it is far weaker and less significant than the correlation with air temperature during the season (see Table 4). There is also no strong correlation between precipitation and discharge more than one season apart. The correlation with the next season is weak and significant only for a few catchments, showing that the

groundwater storage plays an important buffer role. A weak correlation is also seen between winter and the following summer, showing the influence of the remaining snow in summer for a few catchments. Regarding correlation between precipitation and water temperature, only two values are significant for more than 10 catchments. There is a negative correlation between spring precipitation and summer stream temperature, which is discussed below for some particular years. There is also a positive significant correlation for 15 catchments from spring to the following year spring, but since no real physical process explaining

this was found it is assumed to be noise in the results.

Despite this lack of strong correlations on the long-term, connections exist for some single years. Comparison of Figures 11 and 13 shows that a negative relation between spring discharge and summer temperature exists (e.g. 2003 and 2017), but only if the water deficit continues over summer. Indeed, e.g. years 2004, 2005, and 2011 have an important precipitation deficit in spring, without any noticeable above-average water temperature in summer. This means that a spring precipitation deficit can

contribute to a positive summer stream temperature anomaly, but the summer conditions (air temperature and precipitation) remain the main controlling factors and can cancel the spring effect. In fall, impacts of extreme summers as 2003 or 2018 are not noticeable anymore in the mean stream temperature (see Figure 13 and S22).

In summary, from the data sets used in this study, no strong memory patterns could be identified in the hydrological system. While it might be important for more complex systems (e.g. the land-atmosphere interaction), the antecedent state of the system

is not really relevant here and the main drivers of stream temperature remain the current air temperature and to a less extent the discharge.

## 4.5  Ecological indicators

In this section, two ecological indicators based on water temperature are presented. The first one is the number of days per year where the stream temperature exceeds 25°C for at least one hour during the day. Rivers reaching this threshold at least once in

the past are shown in Figure 17. The summer discharge anomaly for these catchments is shown in Figure S24 in supplementary.

There is a noticeable increase in warm water events in the last decades. The extreme years 2003 and 2018 are clearly highlighted. The occurrence of warm days is often related to discharge deficit, i.e. low flow conditions. However, while in the 70's and 80's, peaks above 25°C were only occurring along with a discharge reduction, this is no longer the case in the last decades (see e.g. years 1994, 2007 and 2012), indicating that the Swiss river system is becoming more sensitive and more

exposed to these extreme temperature events with ongoing climate change.

The second threshold is the consecutive number of days above 28 for which the temperature constantly remains above 15°C, which is critical to the spread of the Proliferative Kidney Disease (PKD). Figure 18 shows the number of days per year during which fish would be exposed to PKD. There is a clear increase over the past decades (see Figure 18 bottom panel) and some





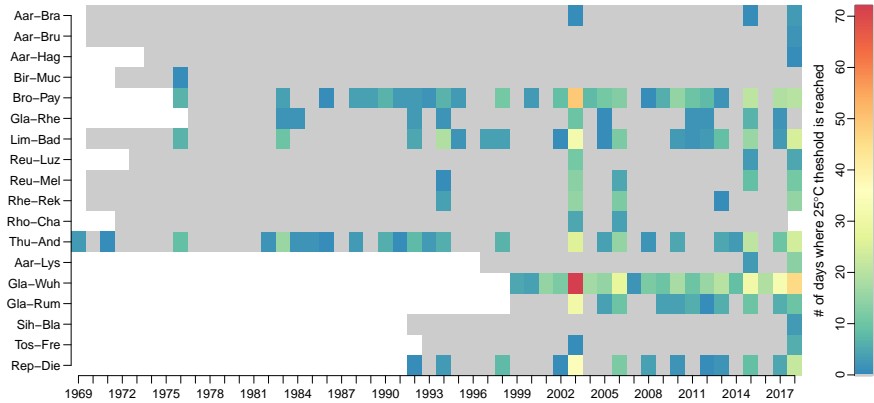

**Figure 17.** Number of days per year when the 25°C threshold is reached (i.e. water temperature is above 25°C for at least 1 hour during the specific day). Only catchments where the threshold is reached at least once are shown. Abbreviations of catchments names are explained in Table 1.

rivers which were almost preserved before 1990, such as the Aare in Bern (Aar-Ber) or the Broye in Payern (Bro-Pay) are more and more affected in the last two decades. During extremes years such as 2003 and 2018, the increase is particularly visible.

Most of the measurement sites where such warm water events were observed are located downstream of lakes and in relatively large catchments, or at low elevation on the Plateau (e.g. the Broye or the Glatt rivers). However, also one small catchment
experiences a large number of days above the threshold (the Alte Aare in Lyss (Aar-Lys), with an area of 13 km$^2$), showing that even small rivers on the Swiss Plateau start to be threatened. Some catchments at higher elevation are also affected (e.g. the Linth in Weesen (Lin-Wee), with a mean basin elevation of 1584 m). Looking at the temporal distribution of days above the 15°C threshold (not shown), they mostly happen between June and mid-October. Over time, there is a clear shift to earlier occurrences in the year, while the ending period remains constant.

**5 Conclusions and outlook**

This detailed analysis of stream temperature trends in Switzerland found strong evidence that climate warming of the last decades had a clear influence on the stream temperature in this largely alpine country. It is in particular also shown that stream temperatures have continued to rise after the shift observed in 1987/1988. For the period 1979-2018 the mean warming rate is +0.33 °C per decade (for the available 31 catchments), and for the period 1999-2018 the mean warming rate is +0.37 °C per
15 decade (considering 52 catchments). This later rate corresponds to about 95 % of the contemporary air temperature warming rate. However, at the single catchment scale, air and water temperature trends are poorly correlated suggesting large influence of local conditions and hydrological processes on water temperature. The warming is more pronounced in summer and less important in winter, creating a gradually increasing winter to summer stream temperature difference. In spring, the water





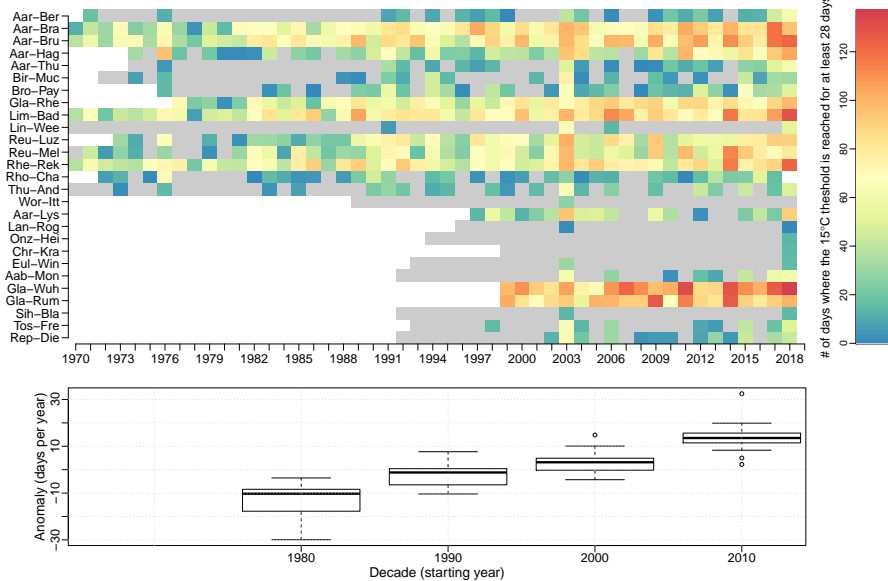

**Figure 18.** Top: Number of days per year when the water temperature is above 15°C since at least 28 days (first 28 days not counted). Abbreviations of catchment names are explained in Table 1. Bottom: Anomaly on the decadal mean number of annual days where the water temperature is above 15°C since at least 28 days (first 28 days not counted) for the 15 catchments where data are available since 1980. Anomaly with respect to the full period mean.

temperature trend is more pronounced than the air temperature trend. While in general the warming of streams is mainly driven by the air temperature, we show that discharge conditions and snow or glacier melt also play an important role, especially in summer. Furthermore, our analysis clearly reveals the role of snow melt in creating resilience to warming in high alpine streams. This resilience is however likely to reduce in the near future due to expected further decreases in future snow cover.

We also show that the presence of lakes speeds up the shift from limited trends in alpine streams to larger ones on the Swiss Plateau, while the catchment area does not have a strong statistical correlation with the observed water temperature trend.

The impact of past climate change on discharge, a key driver of stream temperature, is less clear. A decrease of 10 % per decade is observed over the period 1999-2018. This decrease is more evident in spring and fall while a small increase is observed in winter. The annual discharge evolution is closely related to the annual precipitation evolution. On the longer term,

there are some oscillations in the observed discharge and precipitation time series, and mean discharge similar to today's values were already observed in the past. Therefore, it is not possible yet to assess whether there is a tangible impact from climate change on discharge at the scale of Switzerland.

The relevance of the identified trends for water resources and ecosystem management is underlined by the analysis of temperature threshold exceedance during summer. We show that the legal limit for stream temperature in Switzerland (25°C),

beyond which heat release in any form is prohibited, is reached more often in the past few years and that the conditions for the development of Proliferative Kidney Disease in fish are also met more frequently than in the past. Considering the expected

continuation of air temperature rise in Switzerland (MeteoSuisse et al., 2018), our study shows the urgent need of adaptation and mitigation strategies to preserve the fluvial ecosystems of Switzerland and mitigate the impacts on the Swiss economy and energy production sectors.

While in this study it was attempted to cover and investigate the main hydrological regimes of Switzerland, only five stations for alpine catchments and only one for the southern Alps (Ticino) region have sufficiently long time series for analyses. Indeed, water temperature is a recent and serious concern and the stream temperature measurement networks in the Swiss cantons have mainly been installed after 2000. Based on the present denser network for stream temperature monitoring, and in view of the expected continuation of the temperature rise, it would be interesting to repeat a similar study with the additional available stations in some years from now to detect changes and new trends.

Besides the trend analysis, a key objective of this study was to investigate physical mechanisms underlying stream temperature in different hydrological regimes. The results show that there is no strong memory effect on the system with respect to stream temperature. The water temperature, stream discharge and the meteorological conditions have generally a weak impact on the next season. The strongest effect observed is the impact of a warm and dry spring on the following summer; such a situation is known for impacting the air temperature and then leading to higher water temperature. The importance of the seasonal snow cover and the influence of lakes were also shown to be important factors.

The observation and understanding of such mechanisms are crucial for modelling the evolution of water temperature and discharge in the future. Indeed, most of the current hydrological models are mainly based on statistical empirical relationships and they need to accurately capture the underlying processes to be efficient when forecasting the system evolution using climate change scenarios (Leach and Moore, 2019). In addition, future work using physically based models could help to confirm the mechanisms observed here and their evolution.

*Author contributions.* TEXT

The paper was written by Adrien Michel with contributions from all co-authors. Adrien Michel and Tristan Brauchli collected the data, all authors designed the study. Adrien Michel completed the statistical analysis. All authors gave critical feedback on the manuscript.

*Competing interests.* TEXT

The authors declare that they have no conflict of interest.

*Acknowledgements.* This paper is dedicated to the memory of Olivier Overney. The work was funded by the Swiss Federal Office for the Environment (FOEN), Hydrology Division, CH-3003 Berne under grant no. 15.0003.PJ / Q102-0785. The Office for water and waste (AWA) of Canton Bern, the Office for waste, water, energy and air (AWEL) of Canton Zürich, the Federal Office of Meterology and Climatology





(MeteoSwiss) and the Federal Office for the Environment (FOEN) are greatly acknowledged for the free access to their hydrological and meteorological data. We also acknowledge Love Råman Vinnå and Luca Carraro from EAWAG for helpful discussions about lake temperature and PKD, Tobias Jonas and Nora Helbig from SLF for having provided the snow maps and for constructive discussions, and finally GLAMOS for the glacier data.

All analyses were performed with open and free software (Python and R).

*Code and data availability.*   TEXT

The aim is to share the data set collected in the context of this paper, discussions are still going on with data providers to find an agreement. If successful, the dataset will be made available on EnviDat.ch.

The code will also be available in a well commented and comprehensive form at publication time.

For the time being (review) documented code and data are available at: https://drive.switch.ch/index.php/s/9BEqpBWFkNWoTSU

The main author should be contacted to obtain the password (adrien.michel@epfl.ch).

**Appendix A:  Trends for all hydrometric and meteorological stations**

The annual and seasonal trends for stream temperature and discharge are presented for all catchments in Table A1 over the period 1999-2018 and in Table A2 over the period 1979-2018. The trends for air temperature and precipitation are presented in

Tables S3 and S4 in supplementary. Annual trends are computed with de-seasonalized daily time series while seasonal trends are computed from annual means of each season, meaning that annual and seasonal trends should not be directly compared. This also explains why the standard error on the seasonal trend values is more important than for annual ones (see discussion in Section 3.3).




**Table A1.** Water temperature (left part) and discharge (right part) annual and seasonal trends for all catchments presented in Table 1 over the period 1999-2018. The numbers in brackets indicate the standard error of the computed trends based on linear regression.

| River Name | Water temperature trend (° per decade) | | | | | Discharge trend (% per decade) | | | | |
| --- | --- | --- | --- | --- | --- | --- | --- | --- | --- | --- |
| | Annual | Winter | Spring | Summer | Fall | Annual | Winter | Spring | Summer | Fall |
| Aar-Ber | 0.35 (0.03) | 0.10 (0.19) | 0.20 (0.29) | 0.62 (0.38) | 0.47 (0.21) | -5.9 (0.8) | 5.7 (9.5) | -9.8 (8.9) | -6.9 (4.4) | -7.1 (7.4) |
| Aar-Bri | 0.44 (0.01) | 0.29 (0.12) | 0.17 (0.12) | 0.80 (0.10) | 0.47 (0.08) | -4.6 (0.7) | -4.6 (8.6) | -5.4 (6.2) | -6.1 (2.9) | -1.3 (5.6) |
| Aar-Bru | 0.44 (0.03) | 0.29 (0.30) | 0.32 (0.38) | 0.60 (0.42) | 0.50 (0.24) | -9.4 (0.9) | 6.8 (10.9) | -16.5 (12) | -7.4 (9) | -19.2 (11.3) |
| Aar-bra | 0.44 (0.03) | 0.28 (0.28) | 0.26 (0.34) | 0.69 (0.39) | 0.54 (0.23) | -6.8 (1) | 6.5 (10.8) | -17.8 (14.9) | -0.8 (9.7) | -9.5 (12.4) |
| Aar-hag | 0.58 (0.03) | 0.20 (0.21) | 0.47 (0.37) | 0.92 (0.39) | 0.72 (0.25) | -4.5 (0.9) | 13.5 (10.1) | -10.4 (10.6) | -6.4 (5.9) | -8.1 (8.6) |
| Aar-Rin | 0.25 (0.02) | -0.02 (0.12) | 0.10 (0.22) | 0.61 (0.19) | 0.33 (0.14) | -1.9 (0.7) | -5.6 (7.5) | -10.9 (9.7) | 1.7 (4.4) | 2.4 (7) |
| Aar-Thu | 0.42 (0.03) | 0.19 (0.19) | 0.26 (0.29) | 0.67 (0.38) | 0.56 (0.23) | -7.0 (0.8) | 4.1 (9.9) | -10.8 (8.7) | -7.7 (4) | -8.0 (7.2) |
| Arv-Gva | 0.28 (0.02) | 0.26 (0.28) | 0.23 (0.24) | 0.28 (0.10) | 0.31 (0.13) | -6.1 (1.2) | 11.3 (12.2) | -10.5 (11.5) | -3.7 (7.9) | -18.7 (13.4) |
| Bir-Muc | 0.15 (0.04) | -0.18 (0.32) | 0.08 (0.33) | 0.47 (0.53) | 0.20 (0.20) | -16.9 (2.1) | 4.0 (12.1) | -25.7 (18.1) | -6.1 (19.1) | -43.4 (27.3) |
| Bro-Pay | 0.36 (0.04) | 0.18 (0.35) | 0.33 (0.43) | 0.59 (0.56) | 0.35 (0.27) | -16.9 (2.5) | 6.6 (12.4) | -27.5 (19.3) | -18.4 (21.8) | -31.3 (20.7) |
| Emm-Emm | 0.39 (0.03) | 0.18 (0.22) | 0.45 (0.32) | 0.66 (0.29) | 0.25 (0.25) | -11.9 (2.4) | 4.2 (16) | -16.8 (13.4) | -10.5 (16.4) | -24.3 (20.6) |
| Eaa-Buo | 0.29 (0.02) | 0.27 (0.21) | 0.24 (0.13) | 0.36 (0.12) | 0.25 (0.10) | -0.9 (1.1) | 3.9 (12.1) | -7.7 (9.1) | 0.4 (5.7) | 0.1 (9.5) |
| Gla-Rhe | 0.27 (0.03) | 0.02 (0.27) | 0.27 (0.33) | 0.58 (0.25) | 0.18 (0.22) | -14.7 (1.3) | -0.6 (10.3) | -26.0 (14.7) | -4.3 (13.3) | -28.4 (14.2) |
| Inn-Sch | 0.14 (0.02) | 0.07 (0.12) | 0.03 (0.18) | 0.30 (0.12) | 0.19 (0.14) | -7.8 (0.9) | -12.0 (7) | -5.6 (9.2) | -6.9 (6.4) | -11.5 (10.4) |
| Kem-Emm | 0.66 (0.04) | 0.45 (0.28) | 0.56 (0.40) | 0.98 (0.50) | 0.63 (0.25) | -13.2 (2.3) | 9.6 (14.1) | -17.4 (12.3) | -23.9 (14.9) | -16.0 (18.4) |
| Lim-Bad | 0.37 (0.03) | 0.09 (0.23) | 0.23 (0.38) | 0.65 (0.45) | 0.49 (0.25) | -11.0 (0.9) | 2.1 (9.4) | -18.8 (9.2) | -13.9 (9.2) | -10.7 (12.1) |
| Lin-Mol | 0.38 (0.02) | 0.24 (0.12) | 0.35 (0.14) | 0.51 (0.15) | 0.39 (0.11) | -7.1 (1) | 2.7 (8.9) | -15.0 (8.6) | -10.0 (7.2) | -1.9 (9.1) |
| Lin-Wee | 0.44 (0.03) | 0.13 (0.16) | 0.31 (0.27) | 0.91 (0.45) | 0.43 (0.26) | -9.2 (0.8) | 4.1 (9.3) | -16.7 (8.8) | -12.6 (8.1) | -4.8 (10.1) |
| Lon-Bla | 0.17 (0.01) | -0.09 (0.09) | 0.11 (0.14) | 0.45 (0.11) | 0.20 (0.10) | -6.2 (1.1) | -2.7 (4) | -4.4 (7.4) | -5.4 (4) | -10.7 (7) |
| Lus-Gst | 0.41 (0.02) | 0.14 (0.18) | 0.24 (0.12) | 0.76 (0.09) | 0.47 (0.10) | -3.1 (1) | 2.4 (10.6) | -6.8 (8) | -3.1 (2.9) | -2.0 (7.9) |
| Muo-Ing | 0.14 (0.02) | 0.07 (0.22) | 0.11 (0.16) | 0.19 (0.32) | 0.13 (0.21) | -8.5 (1.5) | 12.4 (13.9) | -15.2 (8) | -13.2 (9.2) | -3.9 (13.5) |
| Reu-Luz | 0.38 (0.03) | 0.18 (0.19) | 0.24 (0.32) | 0.56 (0.40) | 0.51 (0.23) | -7.9 (0.8) | 7.8 (9.2) | -12.9 (7) | -10.5 (5.7) | -7.0 (9.4) |
| Reu-mel | 0.47 (0.03) | 0.29 (0.23) | 0.38 (0.34) | 0.68 (0.42) | 0.50 (0.23) | -6.9 (0.8) | 8.5 (9.9) | -11.1 (7.7) | -9.9 (6.2) | -8.0 (10.6) |
| Reu-See | 0.19 (0.02) | 0.00 (0.14) | -0.02 (0.16) | 0.40 (0.15) | 0.31 (0.14) | -6.4 (0.9) | 4.0 (9.1) | -5.3 (6.7) | -9.0 (4.5) | -7.6 (6.6) |
| Rhe-Die | 0.46 (0.02) | 0.20 (0.16) | 0.29 (0.20) | 0.85 (0.29) | 0.46 (0.20) | -11.0 (0.9) | -1.9 (7.8) | -11.6 (8) | -14.6 (8.3) | -10.9 (9.4) |
| Rhe-Rek | 0.38 (0.03) | 0.12 (0.26) | 0.15 (0.35) | 0.76 (0.48) | 0.52 (0.25) | -12.4 (0.6) | 0.7 (9.2) | -17.5 (8.2) | -14.6 (8.4) | -14.3 (9.2) |
| Rhe-Rhe | 0.51 (0.03) | 0.30 (0.24) | 0.39 (0.38) | 0.76 (0.43) | 0.56 (0.25) | -10.8 (0.7) | 3.1 (9.6) | -17.0 (9.5) | -11.2 (7.6) | -15.5 (10) |
| Rho-Cha | 0.43 (0.04) | 0.43 (0.25) | 0.46 (0.39) | 0.29 (0.41) | 0.64 (0.31) | -8.9 (0.9) | 0.8 (7.1) | -17.2 (11.6) | -5.5 (7) | -15.2 (8.8) |
| Rho-Pds | 0.32 (0.02) | 0.27 (0.13) | 0.30 (0.17) | 0.32 (0.09) | 0.35 (0.12) | -4.3 (0.6) | 7.8 (4) | -7.8 (7.2) | -4.5 (5.3) | -9.1 (3.7) |
| Rho-Sio | 0.13 (0.02) | 0.06 (0.21) | 0.09 (0.19) | 0.17 (0.08) | 0.16 (0.11) | -6.7 (0.8) | -5.7 (4.7) | -5.9 (7.5) | -3.9 (5) | -14.8 (4.9) |
| Thu-And | 0.67 (0.04) | 0.45 (0.30) | 0.60 (0.38) | 1.00 (0.52) | 0.56 (0.25) | -15.8 (2.2) | 2.5 (12.6) | -30.6 (13.6) | -11.2 (13.7) | -20.2 (17.9) |
| Tic-Ria | 0.13 (0.02) | 0.10 (0.19) | -0.05 (0.22) | 0.24 (0.39) | 0.18 (0.20) | -7.5 (1.6) | -4.3 (11.8) | 3.4 (10.6) | -8.0 (12.6) | -20.8 (17) |
| Wor-Itt | 0.24 (0.02) | 0.60 (0.29) | 0.31 (0.24) | -0.03 (0.32) | 0.10 (0.16) | -1.7 (1.2) | 14.1 (11) | -10.8 (14.1) | 4.7 (17.5) | -11.8 (15.5) |
| Kan-Fru | 0.11 (0.02) | 0.07 (0.16) | 0.08 (0.17) | 0.22 (0.15) | 0.06 (0.14) | -5.4 (0.8) | 13.3 (6) | -6.0 (8.9) | -9.3 (4.2) | -2.5 (5) |
| Aar-Lys | 0.28 (0.03) | -0.64 (0.45) | 0.31 (0.44) | 1.13 (0.45) | 0.31 (0.29) | -8.4 (0.2) | -9.0 (1.5) | -7.0 (2.5) | -7.1 (2.6) | -10.3 (1.8) |
| Suz-Vil | 0.23 (0.03) | 0.20 (0.28) | 0.30 (0.21) | 0.12 (0.41) | 0.26 (0.15) | -12.2 (2.5) | 8.8 (13.8) | -25.7 (19.1) | 14.7 (30.3) | -45.3 (25.1) |
| Lan-Rog | 0.58 (0.03) | 0.55 (0.32) | 0.55 (0.25) | 0.83 (0.27) | 0.40 (0.20) | -13.4 (1.2) | 1.5 (8.7) | -19.5 (12.7) | -11.5 (16.3) | -25.4 (13.2) |
| Onz-Hei | 0.41 (0.03) | 0.34 (0.29) | 0.35 (0.24) | 0.63 (0.28) | 0.32 (0.18) | -22.7 (1.4) | -10.7 (9) | -30.7 (13.7) | -18.8 (16.1) | -30.4 (14.4) |
| Osc-Kop | 0.50 (0.03) | 0.48 (0.29) | 0.53 (0.24) | 0.73 (0.29) | 0.30 (0.20) | -6.4 (1.3) | 5.7 (8.7) | -8.5 (13) | -0.6 (15.9) | -21.8 (15.5) |
| Sag-Wor | 0.24 (0.01) | 0.37 (0.11) | 0.13 (0.13) | 0.14 (0.14) | 0.28 (0.10) | 0.1 (2.3) | 6.0 (15.4) | -14.6 (26.3) | 18.7 (25.5) | -5.4 (22.7) |
| Rau-Mou | 0.74 (0.03) | 0.55 (0.35) | 0.64 (0.20) | 0.96 (0.33) | 0.74 (0.19) | -18.7 (2.1) | 3.8 (13.2) | -27.1 (15.5) | -3.2 (21.6) | -46.5 (21.9) |
| Chr-Kra | 0.28 (0.03) | 0.13 (0.36) | 0.15 (0.26) | 0.77 (0.33) | 0.13 (0.20) | -8.3 (1.7) | 11.1 (12.4) | -15.8 (16.4) | -7.0 (19.4) | -19.4 (16.6) |
| Lut-Obe | 0.58 (0.03) | 0.54 (0.35) | 0.56 (0.24) | 0.77 (0.34) | 0.46 (0.23) | -8.2 (1.6) | 13.7 (11) | -24.4 (19.3) | -7.3 (19.8) | -5.1 (14.8) |
| Kem-Ill | 0.38 (0.02) | 0.26 (0.22) | 0.37 (0.27) | 0.46 (0.35) | 0.42 (0.15) | -7.2 (2.7) | 6.5 (14.7) | -19.3 (19) | 10.4 (20.3) | -33.2 (25.6) |
| Tos-Ram | 0.32 (0.03) | 0.37 (0.23) | 0.27 (0.26) | 0.25 (0.21) | 0.42 (0.20) | -17.0 (2.9) | -0.4 (16.2) | -32.0 (16.5) | -2.5 (19.3) | -33.8 (26.8) |
| Eul-Win | 0.33 (0.03) | 0.23 (0.31) | 0.30 (0.31) | 0.52 (0.37) | 0.24 (0.18) | -11.9 (2.3) | -4.3 (14) | -26.9 (17.3) | 1.6 (16.9) | -36.6 (21.8) |
| Aab-Mon | 0.43 (0.03) | 0.23 (0.25) | 0.39 (0.31) | 0.81 (0.37) | 0.27 (0.18) | -13.2 (2.7) | -5.2 (12.5) | -26.7 (17.7) | -1.6 (17.8) | -33.6 (18.6) |
| Gla-Wuh | 0.53 (0.03) | 0.58 (0.26) | 0.40 (0.41) | 0.62 (0.44) | 0.63 (0.26) | -6.5 (1.3) | 8.3 (12.1) | -19.4 (15.7) | 9.4 (16) | -26.4 (17.1) |
| Gla-Rum | 0.32 (0.03) | 0.02 (0.25) | 0.18 (0.39) | 0.66 (0.30) | 0.42 (0.24) | -10.9 (1.4) | -0.1 (11.1) | -20.0 (16.7) | 4.8 (16.8) | -31.9 (15.7) |
| Sih-Bla | 0.40 (0.04) | 0.21 (0.23) | 0.33 (0.34) | 0.50 (0.44) | 0.51 (0.29) | -9.5 (2.5) | 2.2 (10.3) | -24.7 (11.6) | -6.0 (10.4) | -9.4 (11.4) |
| Tos-Fre | 0.53 (0.03) | 0.39 (0.23) | 0.53 (0.35) | 0.66 (0.42) | 0.52 (0.21) | -13.7 (2.1) | -0.9 (12.5) | -24.1 (14.7) | -1.7 (15) | -28.1 (18.9) |
| Rep-Die | 0.38 (0.04) | 0.26 (0.30) | 0.28 (0.36) | 0.60 (0.46) | 0.34 (0.20) | -16.2 (2.5) | -1.4 (12.1) | -25.8 (17.9) | -6.9 (17.4) | -37.3 (19.5) |





**Table A2.** Water temperature (left part) and discharge (right part) annual and seasonal trends for all catchments presented in Table 1 over the period 1979-2018. The numbers in brackets indicate the standard error of the computed trends based on linear regression.

| River Name | Water temperature trend (° per decade) | | | | | Discharge trend (% per decade) | | | | |
|---|---|---|---|---|---|---|---|---|---|---|
| | Annual | Winter | Spring | Summer | Fall | Annual | Winter | Spring | Summer | Fall |
| Aar-Ber | 0.39 (0.01) | 0.16 (0.06) | 0.41 (0.09) | 0.65 (0.12) | 0.32 (0.07) | -1.4 (0.3) | -1.4 (2.9) | 0.9 (2.9) | -2.5 (1.6) | -2.4 (2.4) |
| Aar-Bri | 0.24 (0.01) | -0.02 (0.05) | 0.16 (0.03) | 0.47 (0.05) | 0.31 (0.03) | 0.1 (0.2) | -2.4 (2.3) | 2.7 (2.1) | -0.9 (1.3) | 0.6 (2.1) |
| Aar-Bru | 0.43 (0.01) | 0.27 (0.10) | 0.52 (0.12) | 0.59 (0.13) | 0.33 (0.08) | -4.4 (0.3) | -3.7 (3.6) | -4.0 (3.9) | -3.9 (2.9) | -6.3 (3.4) |
| Aar-bra | 0.43 (0.01) | 0.21 (0.09) | 0.49 (0.12) | 0.65 (0.12) | 0.37 (0.08) | - | - | - | - | - |
| Aar-hag | 0.49 (0.01) | 0.20 (0.07) | 0.48 (0.11) | 0.79 (0.12) | 0.46 (0.09) | - | - | - | - | - |
| Aar-Rin | 0.31 (0.01) | 0.14 (0.05) | 0.43 (0.08) | 0.49 (0.08) | 0.19 (0.06) | -0.5 (0.2) | -4.5 (2.2) | 2.8 (3) | -1.3 (1.4) | 0.1 (2) |
| Aar-Thu | 0.37 (0.01) | 0.17 (0.06) | 0.38 (0.09) | 0.60 (0.12) | 0.32 (0.08) | -1.9 (0.3) | -2.7 (3) | 0.8 (2.9) | -2.8 (1.6) | -3.0 (2.3) |
| Arv-Gva | 0.20 (0.01) | 0.05 (0.09) | 0.25 (0.08) | 0.28 (0.03) | 0.22 (0.04) | -7.5 (0.4) | -5.0 (4.2) | -3.6 (3.6) | -9.7 (2.7) | -12.3 (4.2) |
| Bir-Muc | 0.28 (0.01) | 0.14 (0.11) | 0.36 (0.10) | 0.45 (0.16) | 0.16 (0.07) | -3.5 (0.7) | -0.5 (4.5) | -6.4 (5.6) | -2.8 (6.1) | -2.7 (8.2) |
| Bro-Pay | 0.41 (0.01) | 0.13 (0.12) | 0.51 (0.13) | 0.70 (0.17) | 0.29 (0.09) | -10.6 (0.9) | -7.8 (4.7) | -10.3 (6) | -11.5 (7.1) | -14.0 (6.2) |
| Emm-Emm | 0.35 (0.01) | 0.08 (0.07) | 0.46 (0.10) | 0.60 (0.09) | 0.26 (0.08) | -1.5 (0.8) | -1.6 (5.5) | -4.1 (4.6) | 3.3 (5) | -3.8 (6.4) |
| Eaa-Buo | - | - | - | - | - | -1.5 (0.3) | -3.9 (3.4) | 2.6 (3) | -2.9 (1.9) | -2.0 (3.2) |
| Gla-Rhe | 0.36 (0.01) | 0.19 (0.09) | 0.48 (0.11) | 0.50 (0.08) | 0.24 (0.07) | -5.6 (0.4) | -7.1 (3.7) | -5.8 (4.9) | -2.5 (4.4) | -6.4 (4.8) |
| Inn-Sch | 0.12 (0.01) | 0.04 (0.05) | 0.07 (0.07) | 0.27 (0.04) | 0.12 (0.06) | - | - | - | - | - |
| Kem-Emm | 0.42 (0.01) | 0.20 (0.07) | 0.54 (0.12) | 0.63 (0.15) | 0.31 (0.09) | -3.7 (0.8) | -3.6 (3.9) | -3.0 (4.6) | -5.6 (5.7) |
| Lim-Bad | 0.42 (0.01) | 0.18 (0.07) | 0.49 (0.12) | 0.66 (0.15) | 0.33 (0.09) | -1.5 (0.3) | 0.0 (3) | -0.2 (3.4) | -4.0 (3.2) | -1.0 (4) |
| Lin-Mol | 0.24 (0.01) | 0.14 (0.04) | 0.28 (0.04) | 0.33 (0.05) | 0.19 (0.04) | -2.3 (0.3) | 0.9 (2.7) | -0.4 (3) | -5.8 (2.6) | -1.1 (3.2) |
| Lin-Wee | 0.44 (0.01) | 0.20 (0.05) | 0.44 (0.09) | 0.78 (0.14) | 0.35 (0.09) | -2.5 (0.3) | 0.9 (2.9) | -0.4 (3.1) | -6.6 (2.8) | -0.6 (3.4) |
| Lon-Bla | 0.21 (0.01) | 0.03 (0.04) | 0.18 (0.06) | 0.42 (0.05) | 0.23 (0.04) | -2.7 (0.4) | -1.0 (1.6) | 8.1 (2.8) | -3.1 (1.5) | -8.6 (2.6) |
| Lus-Gst | 0.26 (0.01) | 0.13 (0.07) | 0.23 (0.04) | 0.35 (0.06) | 0.30 (0.04) | -0.8 (0.3) | -0.2 (3.3) | 2.9 (2.9) | -1.7 (1.1) | -2.9 (2.4) |
| Muo-Ing | 0.08 (0.01) | 0.00 (0.07) | 0.05 (0.05) | 0.29 (0.10) | -0.05 (0.08) | -1.4 (0.5) | 2.9 (4.1) | 2.1 (3) | -6.9 (3.1) | 1.3 (4.7) |
| Reu-Luz | 0.48 (0.01) | 0.19 (0.06) | 0.49 (0.10) | 0.81 (0.14) | 0.43 (0.08) | -1.1 (0.3) | 1.4 (2.8) | 2.0 (2.6) | -4.1 (2) | -0.2 (3.2) |
| Reu-mel | 0.43 (0.01) | 0.23 (0.08) | 0.48 (0.10) | 0.65 (0.14) | 0.36 (0.08) | -1.3 (0.3) | -0.5 (3.1) | 1.1 (2.8) | -3.5 (2.2) | -0.9 (3.6) |
| Reu-See | 0.19 (0.01) | -0.07 (0.05) | 0.15 (0.05) | 0.41 (0.04) | 0.23 (0.05) | -2.1 (0.3) | 0.6 (2.6) | 4.2 (2.6) | -5.4 (1.8) | -2.3 (2.8) |
| Rhe-Die | 0.29 (0.01) | 0.10 (0.06) | 0.29 (0.06) | 0.54 (0.09) | 0.22 (0.06) | -3.2 (0.3) | 0.8 (2.3) | 0.4 (2.8) | -7.7 (2.8) | -2.0 (3.4) |
| Rhe-Rek | 0.45 (0.01) | 0.20 (0.09) | 0.49 (0.11) | 0.75 (0.15) | 0.37 (0.09) | -2.5 (0.2) | 0.7 (3) | -0.9 (3.1) | -5.0 (2.8) | -3.0 (3.2) |
| Rhe-Rhe | 0.45 (0.01) | 0.22 (0.09) | 0.50 (0.11) | 0.70 (0.14) | 0.36 (0.09) | -3.3 (0.3) | -1.7 (3.2) | -2.4 (3.3) | -4.6 (2.6) | -4.0 (3.3) |
| Rho-Cha | 0.44 (0.01) | 0.25 (0.08) | 0.52 (0.12) | 0.55 (0.13) | 0.44 (0.10) | -6.7 (0.3) | -4.9 (2.5) | -6.1 (3.5) | -7.1 (2.3) | -8.6 (3) |
| Rho-Pds | 0.24 (0.01) | 0.13 (0.05) | 0.31 (0.05) | 0.21 (0.04) | 0.29 (0.04) | -2.7 (0.2) | 1.2 (1.4) | -1.4 (2.3) | -3.6 (1.8) | -5.2 (1.7) |
| Rho-Sio | 0.13 (0.01) | 0.01 (0.06) | 0.17 (0.06) | 0.15 (0.03) | 0.18 (0.04) | -3.4 (0.3) | -2.1 (1.6) | 0.6 (2.6) | -3.5 (1.8) | -7.4 (2) |
| Thu-And | 0.46 (0.01) | 0.27 (0.10) | 0.53 (0.12) | 0.64 (0.16) | 0.36 (0.09) | -3.9 (0.8) | -3.2 (4.2) | -4.8 (4.6) | -3.4 (4.3) | -4.1 (5.9) |
| Tic-Ria | 0.25 (0.01) | 0.19 (0.06) | 0.20 (0.07) | 0.39 (0.12) | 0.22 (0.07) | - | - | - | - | - |

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
