# Peer review of "Stream temperature and discharge evolution in Switzerland over the last 50 years: annual and seasonal behavior"

_Hydrology and Earth System Sciences, 2019_

## Referee Comment (RC1) · Anonymous Referee #1 · 25 Sep 2019

This is a well written and extensive article that investigates and compares stream temperature, discharge, air temperature, and precipitation trends across Switzerland. In general, I think this is a very interesting historical perspective on how hydrology, weather, and elevation may interact to shape stream temperature responses. On some level, this article almost does too much – there are a lot of analyses! However, I would favor including all of the information as framed, and reducing the number of figures. The article will certinaly add an interesting perspective to the existing stream temperature literature.

Major comments:

-What strikes me in reading this article is that it is billed as a trend assessment of stream temperature, but in reality, the authors have endeavored to characterize trends

in discharge, precipitation, and air temperature as well. With this in mind, I recommend the authors slightly recast their scope and title to indicate the breadth of their analyses. Given so much of the results and discussion are focused on comparing amongst these different trends, I think recasting will only strengthen the manuscript.

-I do appreciate that the authors interpret their stream temperature trend assessment in terms of trends in air temperature, precipitation, and discharge. However, I have two questions and concerns:

First, their interpretation is largely based on data-driven relationships, and not mechanistic relationships. Therefore, inferring correlation means one variable is "driving" or altering the other is not an accurate interpretation. Therefore, I encourage them to revisit some of the statements in their manuscript to more carefully contextualize the responses they see and their interpretation (e.g., pg 15 lines 1 – 5; pg 21, lines 22 - 23)

Second, I'm not sure looking at relationships between annual trends in stream temperature, air temperature, and precipitation are helpful. Would we expect a change in discharge to impact stream temperature, based on first principles? (Even when we know that more water is harder to heat up, if that cold water occurs in a time of the year with limited energy input, does it matter?) What matters much more is when that change occurs, as is described in in the seasonal analysis.

-As someone who thinks a lot about trend analysis of stream temperature, I have found that stream temperature trends can sometimes be driven by outliers, even when using methods that are robust to outliers. For this analysis, are trends robust? If the trends are recomputed with one or two years less of data, do the general trends hold?

-While I like that the article is strongly framed in the context of changes in Switzerland, what I currently feel is missing is a historical perspective on other stream temperature trend assessments. What have others found in the context of historical stream temperature trend analysis? How do the results from this study compare? Broadening the

findings from this specific geographic region would place the study in a larger context, and would add to its impact.

-At current, the article may include too many figures. I'd strongly encourage the authors to reduce the amount of information they show in the main text, and translate more information to Supporting Information.

Minor comments; Pg 3 line 12: "the longer ones" – what does ones refer to? Could you be more specific?

Pg 8 line 7: "which is low for outliers" – I'm not sure what is meant by this

Pg 14 line 1: pluralize "mean"?

Pg 14 line 23 – 24: there's a missing word in here

Pg 17 line 15 – 16: a little awkward – clarifying would help!

Pg 21 line 30 – peculiarity?

Figure 6: It would make the figure more interpretable and cleaner if the figure titles are moved to be labels for the x-axis instead

---

## Referee Comment (RC2) · Jacob Zwart (Referee) · 3 Oct 2019

Michel and co-authors describe in detail stream temperature trends in Switzerland over the last several decades. The authors show how stream temperature trends are modified by catchment type and position on the landscape (e.g. fed by glaciers vs. lake outflow) as well as how seasonal differences (greater difference in winter and summer temperature) underly the annual trends. These trends have important ecological and economic implications as temperature thresholds are being reached more frequently. I think manuscript is well-written and important, however, there are a few items that I think could be clarified before it is suitable for publication. Please see more detailed comments below.

[Figure]

General comment: The manuscript is quite long and I think could be distilled down to a few major points to improve readability. The most important points that came out to me were: 1) water temperature trends are increasing (Figure 2), 2) water temperature trends are influenced by air temperature but also modified by landscape position (I think a modified version of figure 5 would show this well, where water and air trends are plotted against each other and points are colored based on catchment type; Fig 6 also shows this), 3) Seasonal difference underly the annual trends (Fig 8, 15, and 16 show this most strongly), and 4) there are important ecological and economic implications for these temperature trends (fig 17 and 18). I encourage the authors to reduce the number of figures and condense some of the text or move to supplementary to make the main text a little bit more concise.

Specific comments:

Page 1 Line 13: example of ecological temp thresholds?

Page 2 Line 7: what is the global regime shift?

Page 10 Line 7-8: why did the authors choose a 4 hour moving window average?

Figure 2: I like this figure but it is hard to tell which water station site is referring to which line on the top panel. Have the authors considered having the site labels point to the start or end of the 5 year moving average line for each site? This might improve the interpretation of site specific time series, but it also may make the figure too busy. Another option would be to order the site abbreviations in order of stream temperature from end of 5 year moving average line (i.e. year 2016) rather than what appears to be alphabetically-ordered currently.

Figure 3. I think the labels could be ordered differently or point to the lines to which they correspond. See my comment for Figure 2 above.

Figure 5. Rather than plotting boxplots next to each other, I think plotting the water temperature trends vs. air temperature trends as well as discharge trends vs precipita-

[Figure]

tion trends would convey more information. These scatter plots could also be colored by stream regime. You could keep the boxplots as marginal plots on the scatter plot figure to retain quartile and median information. It would be interesting to see when water and air temperature trends are correlated and when they are not.

Page 13 into Page 14: "Indeed, for both pairs, the hypothesis of different mean is clearly rejected with p-values>0.15." Is this testing the difference between DLA and SPJ, and ALP and HYP? I assume so, but I think this could be written a little bit more clearly to make explicit.

Page 15 lines 12-13: It isn't clear to me how the authors concluded that air temperature is this main driver of water temperature trends for the SPJ catchments. Figure 5 shows a comparison between water and air temperature trends but all of the catchment types are grouped together in this figure so it is impossible to see the effect of air temp on water temp for SPJ catchments specifically. Please be clearer as to how you came to this conclusion.

Page 15 line 20: include some citations for the statement that this is 'well known'.

Figure 7: I suggest adding the label 'inflow' and 'outflow' to the figure itself to help the reader quickly understand the figure rather than having to read through the legend to understand which line is inflow and which is outflow.

Page 18 line 4: Is 'intra-annual' not 'infra-annual' more appropriate here?

Figure 8: indicate what the panel months mean (DJF, JJA, etc...). I was confused until I read the main text.

---

## Author Comment (AC1) · 3 Oct 2019

Dear reviewer,

First of all, we would like to thank the reviewer for his/her clear and relevant review. We provide our detailed answers below.

Review:

"What strikes me in reading this article is that it is billed as a trend assessment of stream temperature, but in reality, the authors have endeavored to characterize trends in discharge, precipitation, and air temperature as well. With this in mind, I recommend the authors slightly recast their scope and title to indicate the breadth of their analyses. Given so much of the results and discussion are focused on comparing amongst these

different trends, I think recasting will only strengthen the manuscript."

Answer:

Indeed, our analysis reaches further than stream temperature trends alone. This comes quite naturally, since discharge (Q) and stream temperature (T) are inherently linked to precipitation (P) and air temperature (TA). Isolated analysis of stream temperature does not give the full picture. The current title was motivated by the research focus of the underlying research project. We agree that a broader title would be beneficial and appropriate. An adapted title could be: "Understanding stream temperature and discharge evolution in Switzerland over the last 50 years: annual and seasonal behavior". In addition, in a revised version, we will emphasize in the introduction that stream temperature studies are intrinsically linked to discharge, and thus precipitations.

Review:

"First, their interpretation is largely based on data-driven relationships, and not mechanistic relationships. Therefore, inferring correlation means one variable is "driving" or altering the other is not an accurate interpretation. Therefore, I encourage them to revisit some of the statements in their manuscript to more carefully contextualize the responses they see and their interpretation (e.g., pg 15 lines 1 – 5; pg 21, lines 22 -23)."

Answer:

Thanks for this pertinent comment. Part of the analysis is based on correlations between the considered key variables. We agree that without underlying physical basis to infer causality, statements should only be about observed correlations. In the revision of the manuscript, we will either provide physical justification of the statements, which is easily feasible for statements such as in p.21 l.22-23, or alternatively statements will be changed to only describe the observed correlation, which is totally acceptable since the main goal of the paper is not to infer the physical processes.

Review:

"Second, I'm not sure looking at relationships between annual trends in stream temperature, air temperature, and precipitation are helpful. Would we expect a change in discharge to impact stream temperature, based on first principles? (Even when we know that more water is harder to heat up, if that cold water occurs in a time of the year with limited energy input, does it matter?) What matters much more is when that change occurs, as is described in in the seasonal analysis."

Answer:

Sections 4.1 and 4.2 are meant to present the annual trends of the four discussed variables. We compare the evolution of air and water temperature on the one hand and of discharge and precipitation on the other hand. We think it is interesting to quantify how strong these relationships are and how meteorological variables can be used as proxy for the annual evolution of the hydrological variables (Q and Tw). But we agree that it is indeed somewhat speculative to discuss the impact of discharge/precipitation on stream temperature on an annual basis. Since this is clearly and more accurately discussed in the sections on the seasonal analysis, parts of Sections 4.1 and 4.2 will be rephrased or removed to avoid any confusion w.r.t. to the point raised by the reviewer (e.g. p.11 lines 5 to 11).

Review:

"As someone who thinks a lot about trend analysis of stream temperature, I have found that stream temperature trends can sometimes be driven by outliers, even when using methods that are robust to outliers. For this analysis, are trends robust? If the trends are recomputed with one or two years less of data, do the general trends hold?"

Answer:

Trends are computed on an annual basis, using de-seasonalized daily time series, and on seasonal basis using seasonal means. These two analyses use a number of points
differing by two orders of magnitude.

Regarding seasonal trends, we state in Section 3.3, in particular on p.9 l. 23-24, that seasonal trends are not robust. The calculated seasonal trends are used only in Figure 10, where actual values are used for a rather qualitative analysis. In addition, on p. 20 l. 3-5 we say: "This absence of correlation results from the noise in the individual trend values due to the relatively short time series available. This is a limitation of the applied method and thus trends cannot be used for an inter-variable interaction study." We thus believe that the original text for seasonal trend analysis does not need modification.

Regarding the annual trend analysis, the robustness can indeed also be questioned and we definitely want to address this question in the revised paper. In particular since the main results from the annual trend analysis are presented in the abstract and the conclusion. In response to this reviewer's comment, we propose using two methods for testing the robustness of the trends. The first method is, as proposed by the reviewer, to remove one year at the beginning of the period or one year at the end. Trends using these shortened periods are compared to trends over the full period. The results of this analysis are shown in Figure 1 and 2 at the end of the present response.

Figure 1 shows the analysis for the period 1999-2018. The trends for water and air temperature are indeed lower when the last year 2018 (which was extremely warm in Switzerland) is removed, while for discharge and precipitation the negative trends are less pronounced when the first year 1999 is removed. These differences are notable, but do not change the main message of the study. For the period 1979-2018, removing one year, both at the beginning or at the end of the time interval, leads to almost negligible difference, showing the overall high robustness of the trends over 40 years.

A second approach is to use a robust linear model method (Hampel, 1986) which is implemented in the "rlm" function from the MASS package in R (see https://www.rdocumentation.org/packages/MASS/versions/7.3-51.4/topics/rlm for details). This method aims at producing trends less sensitive to outliers. While it is well

suited for temperature de-seasonalized time series, this method has an issue when coping with the remaining variability in the de-seasonalized discharge and precipitation time series. It even fails to converge for the precipitation time series.

Figure 3 and 4 show the differences in trends obtained from normal and robust linear model methods for the four variables. The only notable difference is for discharge during the period 1999-2018. However, the observed difference is smaller than in the first analysis using the shortened time periods.

As a result of this robustness analysis based on two independent methods, we conclude that the trends for the period 1979-2018 are robust. Regarding the trends over the shorter periods, the main message of the paper is not influenced by the result of this analysis. Nevertheless, we intend to indicate the uncertainty on the trend values in a revised version. We are aware that 20 years is a rather short time period for statistical analysis. However, as explained in the manuscript, many stations have been installed only at the end of the 20th century. It would definitely be worth reproducing such and analysis every ten years using corresponding extended data sets. We propose this study as a first assessment, with time series just long enough to be significant (note that many other stations have been installed after 2000 and therefore have not been used in this study because time series are too short for being significant).

Practically, here is what we propose to include in a revised manuscript:

- Explain the potential influence of outliers and boundary values in Section 3.3

- Explain in the same section the double robustness analysis performed here, adding the four figures in Supplementary.

- Use the maximum difference between trends over the full period and trends with one year removed at the beginning or at the end as an indication of the trend values uncertainty. This would be indicated e.g. in Tables A1, A2, S3 and S4. The current uncertainty indicated in the table is the uncertainty obtained from the linear model
computation, but it is indeed underestimated. The uncertainty on the mean values of the trends can be obtained from the RMSE as shown in the plots.

- Add the robustness analysis to the R package provided together with the paper.

Review:

"While I like that the article is strongly framed in the context of changes in Switzerland, what I currently feel is missing is a historical perspective on other stream temperature trend assessments. What have others found in the context of historical stream temperature trend analysis? How do the results from this study compare? Broadening the findings from this specific geographic region would place the study in a larger context, and would add to its impact."

Answer:

Undoubtedly, this study focuses exclusively on Switzerland because it covers many different hydrological regimes and long historical records are available. Moreover, the present study is part of the boarder project HYDRO-CH2018, which aims at assessing the impact of climate change on the Swiss hydrological system in a wide sense (see https://www.nccs.admin.ch/nccs/en/home/the-nccs/priority-themes/hydro-ch2018/hydro-ch2018-forschungsprojekte.html).

We agree that our results could benefit not only to Switzerland but to a wider community. Comparison with results found in other locations is obviously relevant and of interest, both in terms of trends and in terms of correlations between variables and identified underlying physical processes. Some references to studies in other regions are given in the Introduction p.2 l. 3-4. We take up the reviewer's remark and suggestion and plan to extend this paragraph in the revised version to inform the reader on the main findings of those studies, extend the list of studies presented (a few new papers have been published since), and comparison and discussion will be added in the Conclusion section when and where it is relevant.

Review:

"At current, the article may include too many figures. I'd strongly encourage the authors to reduce the amount of information they show in the main text, and translate more information to Supporting Information."

Answer:

We are aware of it and already tried to reduce the length of the manuscript. Based on all reviewer comments, we will decide how to shorten the manuscript. Any suggestion would be welcome.

Minor comments:

Thanks for these comments; all of them will be addressed in the revised version.

We thank again the reviewer for the constructive and useful comments.

Adrien Michel, on behalf of all authors.

References:

F. R. Hampel, E. M. Ronchetti, P. J. Rousseeuw and W. A. Stahel (1986) Robust Statistics: The Approach based on Influence Functions. Wiley.

[Figure]

**Period 1999–2018, normal linear regression**

**T**

Period 2000–2018, RMSE: 0.058
Period 1999–2017, RMSE: 0.108
Sample mean

**TA**

Period 2000–2018, RMSE: 0.025
Period 1999–2017, RMSE: 0.14
Sample mean

**Q**

Period 2000–2018, RMSE: 4.643
Period 1999–2017, RMSE: 2.432
Sample mean

**P**

Period 2000–2018, RMSE: 3.43
Period 1999–2017, RMSE: 1.379
Sample mean

**Fig. 1.** Comparison between trends over the full period and between trends obtained by re-
moving on year at the beginning (red dots) or at the end of the period (green dots). Period
1999-2018.

[Figure]

**Fig. 2.** Comparison between trends over the full period and between trends obtained by removing on year at the beginning (red dots) or at the end of the period (green dots). Period 1979-2018.

[Figure]

**Fig. 3.** Comparison between trends obtained using a standard linear model and a robust linear model. Period 1999-2018.

**HESSD**

[Figure]

**Fig. 4.** Comparison between trends obtained using a standard linear model and a robust linear model. Period 1979-2018.

[Figure]

---

## Author Comment (AC2) · 11 Oct 2019

Dear Jacob Zwart,

First of all, we would like to thank you for this clear, constructive and helpful review. We provide our detailed answers and explanations below.

Review:

"The manuscript is quite long and I think could be distilled down to a few major points to improve readability. The most important points that came out to me were: 1) water temperature trends are increasing (Figure 2), 2) water temperature trends are influenced by air temperature but also modified by landscape position (I think a modified version of figure 5 would show this well, where water and air trends are plotted against each

other and points are colored based on catchment type; Fig 6 also shows this), 3) Seasonal difference underly the annual trends (Fig 8, 15, and 16 show this most strongly), and 4) there are important ecological and economic implications for these temperature trends (fig 17 and 18). I encourage the authors to reduce the number of figures and condense some of the text or move to supplementary to make the main text a little bit more concise."

Answer:

We are aware of this (it was also pointed out by the other reviewer) and we agree that some content could be moved to supplementary condensing the main findings of the paper. We take note of what you highlighted to be the main figures of the paper and will move some of the secondary figures to the Supplementary in a revised version. The suggestion regarding revision of Figure 5 is discussed further below.

Specific comments:

"Page 1 Line 13: example of ecological temp thresholds?"

These thresholds are for example the 15°C for the PKD spread impacting salmonid fish. We will change this sentence indicating "ecologically and economically relevant temperature thresholds", linking it to our investigation of the legally imposed threshold of 25°C for water usage for industrial cooling in Switzerland (most importantly for nuclear power plants).

"Page 2 Line 7: what is the global regime shift?"

Thanks for the question. It refers to a step change from the 1970s to the 1980s affecting climate and ecosystems mainly in central Europe but has been noticed also in other parts of the world. A good description can be found in Serra-Maluquer (2018):

"In the last four decades, a warming trend has been observed in the Iberian Peninsula; particularly, a rapid rise in temperatures has occurred since the 1980s followed by successive severe droughts in the 1990s, 2000s and 2010s (Gonzalez-Hidalgo and

others, 2015). Such abrupt warming occurred in the transition from the 1970s to the 1980s, and it was partly linked to changes in the winter atmospheric circulation over the northern Atlantic Ocean (Hurrell, 1996) and impacted ecosystems worldwide by accelerating climate warming (Reid and others, 2016). This climate shift has led to warmer and more arid conditions on several European regions, generating harsher climatic conditions for beech forests".

The appropriate references will be added in the revised version.

"Page 10 Line 7-8: why did the authors choose a 4 hours moving window average?"

The moving average is used to smooth the data avoiding to discard periods when the measured temperature was below 15°C just for one hour only. We also realize that the value of 4 hours was a typo, the actual value used is 3 (indeed moving window size for smoothing is always odd); this will be corrected in the revised version.

We made a sensitivity test on the value of the window size and on the total length of the periods while we developed this simple model. Results showed that the chosen values have little impact on the results (see examples in the additional Figures 1-3 in the present document).

"Figure 2: I like this figure but it is hard to tell which water station site is referring to which line on the top panel. Have the authors considered having the site labels point to the start or end of the 5 year moving average line for each site? This might improve the interpretation of site specific time series, but it also may make the figure too busy. Another option would be to order the site abbreviations in order of stream temperature from end of 5 year moving average line (i.e. year 2016) rather than what appears to be alphabetically-ordered currently."

Thank you for the suggestion; please see in Figure 4 a proposition for an updated version considering your comments. In our opinion, the best solution is ordering the site abbreviations in terms of stream temperature values at the end of the 5-year moving

average period (as you suggested). This ordering will be explained in the revised figure caption. Adding labels to the plot, or even only numbers, resulted in too much confusion and was not retained as a satisfying solution.

"Figure 3. I think the labels could be ordered differently or point to the lines to which they correspond. See my comment for Figure 2 above. "

Absolutely. We modified analog to Figure 2 (see Figure 5 in the present document for a new version). We also realized that values for the Alte-Aare were missing in the plot (off-scale); we now divided these values by 4 to fit in the plot, and explained it in the caption of the figure. This high value for the specific discharge arises because the Alte Aare has a small proper catchment size (13 km2), but has been artificially connected to the Aare. As a consequence, the discharge is far higher than the discharge we would expect from a catchment that small (see https://hydromaps.ch/#en/13/47.0536/7.2962/bl_hds).

"Figure 5. Rather than plotting boxplots next to each other, I think plotting the water temperature trends vs. air temperature trends as well as discharge trends vs precipitation trends would convey more information. These scatter plots could also be colored by stream regime. You could keep the boxplots as marginal plots on the scatter plot figure to retain quartile and median information. It would be interesting to see when water and air temperature trends are correlated and when they are not."

Thanks for this suggestion. Our original version of this figure was exactly what you proposed, see Figure 6 in the present document. As mentioned in the manuscript, there is far more noise in the water temperature trends than in the air temperature trends. Related to this, see details in our reply to reviewer one (pages C3-4), there is non-negligible uncertainty around the trend values. For these reasons, we decided to present the results with boxplots, which implies a statistical preprocessing allowing to better visualize the signal in the noise. For these reasons we prefer to keep the figures in the current version.

We are not sure whether Figure 6 will make the situation clearer. However, this Figure could be added in the Supplementary.

Also, when looking at Figure 6, one could question the point in the top left corner. This point is a trend value for the water gauging station Rauss/Moutier, with meteorological values from Delemont. While the long-term trend in air temperature seems clear (0.43°C per decade for the period 1979-2018 and 0.46°C per decade for the period 1970-2018 obtained with the linear model for this station), no trend in air temperature is found for the last 20 years, explaining the unexpected position of this point. This is a good example of the noise obtained from a simple linear regression analysis (see more details about trend robustness on the answerer to reviewer one).

"Page 13 into Page 14: "Indeed, for both pairs, the hypothesis of different mean is clearly rejected with p-values>0.15." Is this testing the difference between DLA and SPJ, and ALP and HYP? I assume so, but I think this could be written a little bit more clearly to make explicit."

Yes, here we refer to the values in Table-2, where each pair of regime trends is tested against each other to infer whether the means are similar or not. In a revised version we will rephrase this paragraph to make this point clearer to the reader.

"Page 15 lines 12-13: It isn't clear to me how the authors concluded that air temperature is this main driver of water temperature trends for the SPJ catchments. Figure 5 shows a comparison between water and air temperature trends but all of the catchment types are grouped together in this figure so it is impossible to see the effect of air temp on water temp for SPJ catchments specifically. Please be clearer as to how you came to this conclusion. "

Indeed, the required Figure to illustrate this statement is for now missing from the paper. A Figure corresponding to Figure 6, but for air temperature and precipitations, will be added in the supplementary. This new figure is shown in Figure 7 here below. By comparing the top-left panels of Figure 6 in the manuscript and Figure 7 here, we

clearly see that SPJ water temperature trends are, on average, close to the air temperature trends, which is not the case for HYP and ALP catchments (DLA catchments are discussed in Section 4.3). In Figure 7 in the present document we can also see that water temperature trends are spread around air temperature trends values for SPJ catchments, while they are systematically below for HYP and ALP below.

Nevertheless, we agree that the paragraph on p.15 lines 12-16 needs revision to clearly state that we talk about the mean behavior of the trend and not about single catchments. In addition, comments related to the noise in the single trend comparisons will be added. While part of this noise is caused by the method and the choice of the meteorological stations for the catchments, this noise shows that at the single catchment scale, independently of the regime, many factors other than the air temperature seem to be important. This will also be mentioned in a revised version. Please see our reply to reviewer 1 (page C3) for more detail.

"Page 15 line 20: include some citations for the statement that this is 'well known'."

We can cite for instance Råman Vinnå (2018) and Webb (2007). These references will be added in the revised manuscript.

"Figure 7: I suggest adding the label 'inflow' and 'outflow' to the figure itself to help the reader quickly understand the figure rather than having to read through the legend to understand which line is inflow and which is outflow."

Modified as suggested (see revised Figure 7 here as Figure 8). Indeed, this eases reading of the figure. Additionally, to avoid confusion, we changed the colors in the bottom panel since the meteorological stations do not necessarily match with the rivers shown (the number of available water temperature and meteorological stations can differ).

"Page 18 line 4: Is 'intra-annual' not 'infra-annual' more appropriate here?"

Good catch. Indeed, this is a language mistake, "infra" meaning "below" while "intra" means "within". 'Infra-annual' will be replaced by 'intra-annual' in the whole text. Thanks for pointing this out.

"Figure 8: indicate what the panel months mean (DJF, JJA, etc...). I was confused until read the main text."

The letters refer to the initials of the three months of a season defined as December-January-February (DJF) and so on. This information will be added in the caption of the figure.

We thank again the reviewer for the useful and pertinent comments and for taking the time to go through this comprehensive manuscript.

Adrien Michel, on behalf of all authors.

References:

Gonzalez-Hidalgo JC, Peña-Angulo D, Brunetti M, Cortesi N. 2015. Recent trend in temperature evolution in Spanish mainland (1951–2010): from warming to hiatus. Int J Climatol 36:2405–16. https://doi.org/10.1002/joc.4519.

Hurrell JW. 1996. Influence of variations in extratropical wintertime teleconnections on Northern Hemisphere temperature. Geophys Res Lett 23:665–8

Råman Vinnå, L., Wüest, A., Zappa, M., Fink, G., & Bouffard, D. (2018). Tributaries affect the thermal response of lakes to climate change. Hydrology & Earth System Sciences, 22(1)

Reid PC, Hari RE, Beaugrand G, Livingstone DM, Marty C, Straile D, Barichivich J, Goberville E, Adrian R, Aono Y et al. 2016. Global impacts of the 1980s regime shift. Glob Change Biol 22:682–703. https://doi.org/10.1111/gcb.13106. Serra-Maluquer, X., Gazol, A., Sangüesa-Barreda, G. et al. 2019. Geographically Structured Growth decline of Rear-Edge Iberian Fagus sylvatica Forests After the 1980s Shift Toward a Warmer Climate. Ecosystems 22: 1325. https://doi.org/10.1007/s10021-019-00339-

z Webb B. W. & Nobilis F. (2007). Long-term changes in river temperature and the influence of climatic and hydrological factors, Hydrological Sciences Journal, 52:1, 74-85, DOI: 10.1623/hysj.52.1.74

[Figure]

[Figure]

**Fig. 1.** Sames as Figure 18 in the paper, but without moving window average.

[Figure]

Fig. 2. Sames as Figure 18 in the paper, but with 5 hours moving window average.

[Figure]

Fig. 3. Sames as Figure 18 in the paper, but with 7 hours moving window average.

**Fig. 4.** New proposition for Figure 2 (legend orderred by stream temperature values at the end of the moving window average time series).

[Figure]

[Figure]

**Fig. 5.** New proposition for Figure 3 (legend orderred by discharge values at the end of the moving window average time series).

**Annual T trend as function of annual TA trend**

**Fig. 6.** Scatter plot of air temperature trends (TA) and water temperature trends (T).

[Figure]

**Fig. 7.** Same figure as Figure 6, but for air temprature and precipitation, to be added in Supplementary.

[Figure]

[Figure]

**Fig. 8.** New proposition for figure 7.

---

## Author Response (AR1)

Dear Editor, Dear Reviewers,

Please find below the detailed answer to all the provided comments and questions. Pages and line numbers refer to the track change version of the manuscript contained in the present document, except when document is explicitly indicated. The track changed supplementary are also part of this document.

**Reviewer 1**

Review:

*"What strikes me in reading this article is that it is billed as a trend assessment of stream temperature, but in reality, the authors have endeavored to characterize trends in discharge, precipitation, and air temperature as well. With this in mind, I recommend the authors slightly recast their scope and title to indicate the breadth of their analyses. Given so much of the results and discussion are focused on comparing amongst these different trends, I think recasting will only strengthen the manuscript."*

Answer:

Indeed, our analysis reaches further than stream temperature trends alone. This comes quite naturally, since discharge (Q) and stream temperature (T) are inherently linked to precipitation (P) and air temperature (TA). Isolated analysis of stream temperature does not give the full picture. We agree that better reflects this point and propose an extension in the revised version. The new title reads as:

*"Stream temperature and discharge evolution in Switzerland over the last 50 years: annual and seasonal behavior"*

In addition, in the revised version we emphasize in the introduction that stream temperature studies are intrinsically linked to discharge, and thus precipitations.

Review:

*"First, their interpretation is largely based on data-driven relationships, and not mechanistic relationships. Therefore, inferring correlation means one variable is "driving" or altering the other is not an accurate interpretation. Therefore, I encourage them to revisit some of the statements in their manuscript to more carefully contextualize the responses they see and their interpretation (e.g., pg 15 lines 1 – 5; pg 21, lines 22 -23)."*

Answer:

Thanks for this pertinent comment. Part of the analysis is based on correlations between the considered key variables. We agree that without underlying physical basis to infer causality, statements should only be about observed correlations. Accordingly, several statements have been removed or modified (see page 15 line 9 and lines 14–20 , page 16 lines 5–8, page 19 lines 3–5, and page 23 line 4 in the track-changed version).

Review:

*"Second, I'm not sure looking at relationships between annual trends in stream temperature, air temperature, and precipitation are helpful. Would we expect a change in discharge to impact stream temperature, based on first principles? (Even when we know that more water is harder to heat up, if that cold water occurs in a time of the year with limited energy input, does it matter?) What matters much more is when that change occurs, as is described in in the seasonal analysis."*

Answer:

Sections 4.1 and 4.2 are meant to present the annual trends of the four discussed variables. We compare the evolution of air and water temperature on the one hand and of discharge and precipitation on the other hand. We think it is interesting to quantify how strong these relationships are and how meteorological variables can be used as proxy for the annual evolution of the hydrological variables (Q and Tw). But we agree that it is indeed somewhat speculative to discuss the impact of discharge/precipitation on stream temperature on an annual basis. Since this is clearly and more accurately discussed in the sections on the seasonal analysis, parts of Sections 4.1 and 4.2 have been be rephrased or removed to avoid any confusion w.r.t. to the point raised by the reviewer (see page 13 lines 2–5 and 7–8, page 15 lines 17–20).

Review:

*"As someone who thinks a lot about trend analysis of stream temperature, I have found that stream temperature trends can sometimes be driven by outliers, even when using methods that are robust to outliers. For this analysis, are trends robust? If the trends are recomputed with one or two years less of data, do the general trends hold?"*

Answer:

Trends are computed on an annual basis, using de-seasonalized daily time series, and on seasonal basis using seasonal means.

Regarding seasonal trends, we state in Section 3.3 that seasonal trends are not robust (indeed, these use only few tens of points compared to few thousands for annual trends based on daily values). The calculated seasonal trends are used only in Figure 10, where actual values are used for a rather qualitative analysis. In addition, on p. 21 l. 5-7 we say: "This absence of correlation results from the noise in the individual trend values due to the relatively short time series available. This is a limitation of the applied method and thus trends cannot be used for an inter-variable interaction study." We thus believe that the original text for seasonal trend analysis does not need modification.

Regarding the annual trend analysis, the robustness can indeed also be questioned and has to be addressed. In particular since the main results from the annual trend analysis are presented in the abstract and the conclusion.

In response to this reviewer's comment, we propose using two methods for testing the robustness of the trends. These methods are described in the revised manuscript in Sections 3.3 and Section S1.3 in supplementary.

The first method is, as proposed by the reviewer, to remove one year at the beginning of the period or one year at the end. Trends using these shortened periods are compared to trends over the full period.

Figure S11 in supplementary shows the analysis for the period 1999-2018. The trends for water and air temperature are indeed lower when the last year 2018 (which was extremely warm in Switzerland) is removed, while for discharge and precipitation the negative trends are less pronounced when the first year 1999 is removed. These differences are notable, but do not change the main message of the study. For the period 1979-2018 (see Figure S12 in supplementary), removing one year, both at the beginning or at the end of the time interval, leads to almost negligible difference, showing the overall high robustness of the trends over 40 years. Removing two years instead of one lead to similar results.

A second approach is to use a robust linear model method (Hampel, 1986) which is implemented in the "rlm" function from the MASS package in R (see https://www.rdocumentation.org/packages/MASS/versions/7.3-51.4/topics/rlm for details). This method aims at producing trends less sensitive to outliers. While it is well suited for temperature de-seasonalized time series, this method has an issue when coping with the remaining variability in the de-seasonalized discharge and precipitation time series. It even fails to converge for the precipitation time series.

Figures S9 and S10 in supplementary show the differences in trends obtained from normal and robust linear model methods for the four variables. The only notable difference is for discharge during the period 1999-2018. However, the observed difference is smaller than in the first analysis using the shortened time periods (see above).

As a result of this robustness analysis based on two independent methods, we conclude that the trends for the period 1979-2018 are robust. Regarding the trends over the shorter periods, the main message of the paper is not influenced by the result of this analysis. Nevertheless, we indicate the uncertainty on the trend values in the revised version. We are aware that 20 years is a rather short time period for statistical analysis. However, as explained in the manuscript, many stations have been installed only at the end of the 20th century. It would definitely be worth reproducing such and analysis every ten years using corresponding extended data sets. We propose this study as a first assessment, with time series just long enough to be significant (note that many other stations have been installed after 2000 and therefore have not been used in this study because time series are too short for being significant).

The maximum difference between trends over the full period and trends with one year removed at the beginning or at the end are now used as an indicatior of the trend values uncertainty. This has been added to Tables A1, A2, S3 and S4. The uncertainty on the mean values of the trends can be obtained from the RMSE as shown in the plots and is now added to the values given in the text.

Review:

*"While I like that the article is strongly framed in the context of changes in Switzerland, what I currently feel is missing is a historical perspective on other stream temperature trend assessments. What have others found in the context of historical stream temperature trend analysis? How do the results from this study compare? Broadening the findings from this specific geographic region would place the study in a larger context, and would add to its impact."*

Answer:

Undoubtedly, this study focuses exclusively on Switzerland because it covers many different hydrological regimes and long historical records are available. Moreover, the present study is part of the broarder project HYDRO-CH2018, which aims at assessing the impact of climate change on the Swiss hydrological system in a wide sense (see

https://www.nccs.admin.ch/nccs/en/home/the-nccs/priority-themes/hydro-ch2018/hydro-ch2018-forschungsprojekte.html).

We agree that our results could benefit not only to Switzerland but to a wider community. Comparison with results found in other locations is obviously relevant and of interest, both in terms of trends and in terms of correlations between variables and identified underlying physical processes. Some references to studies in other regions are given in the first paragraph of the Introduction. We took up the reviewer's remark and expanded this paragraph in the revised version to inform the reader on the main findings of these studies; we extended the list of studies presented, and some related comparison and discussion have been added in the Conclusion section where relevant (see page 2 line12–19 and page 29 lines 10–24).

Review:

*"At current, the article may include too many figures. I'd strongly encourage the authors to reduce the amount of information they show in the main text, and translate more information to Supporting Information."*

Answer:

This problem was pointed out by both reviewers and we agree that the article is quite long. As a result, we modified it as follows:

- Table 2, Figure 11 and 12 (in the first manuscript version) have been moved to Supplementary
- Figure 12 (14 in first manuscript version) has been slightly modified and moved to Supplementary.
- Sections 4.4.5 and 4.4.6 have been merged and reduced, part of the content has been moved to Supplementary. Figures 14 and 16 and Table 5 have been moved to Supplementary.

Despite the addition of some content in the introduction and methods section (following reviewers' suggestion), these changes allowed to reduce by 30% the number of figures, by 40 % the number of tables, and by 10% the total length of the paper.

Minor comments:

*Pg 3 line 12: "the longer ones" – what does ones refer to? Could you be more specific?*

This was meaning "the longest times series available", but is has been removed because it is not adding any information and years indicated where for temperature while times series for discharge are available (and used in the paper), since the early 20st century.

*Pg 8 line 7: "which is low for outliers" – I'm not sure what is meant by this*

This has been clarified in the reviewed version (see page 8 lines 20–24).

*Pg 14 line 1: pluralize "mean"?*

Thank you for the catch, it has been corrected.

*Pg 15 line 23 – 24: there's a missing word in here*

Thank you for the catch, it has been corrected.

*Pg 17 line 15 – 16: a little awkward – clarifying would help!*

This has been clarified in the revised version (see page 19 lines 1–15).

*Pg 21 line 30 – peculiarity?*

Thank you, this comes from a confusion with similar a word in French, the statement has been corrected.

*Figure 6: It would make the figure more interpretable and cleaner if the figure titles are moved to be labels for the x-axis instead*

The figure has been reworked in the revised version. Thank you, it looks better and is easier to read now.

**Reviewer 2**

Review:

*"The manuscript is quite long and I think could be distilled down to a few major points to improve readability. The most important points that came out tome were: 1) water temperature trends are increasing (Figure 2), 2) water temperature trends are influenced by air temperature but also modified by landscape position (I think a modified version of figure 5 would show this well, where water and air trends are plotted against each other and points are colored based on catchment type; Fig 6 also shows this), 3) Seasonal difference underly the annual trends (Fig 8, 15, and 16 show this most strongly), and 4) there are important ecological and economic implications for these temperature trends (fig 17 and 18). I encourage the authors to reduce the number of figures and condense some of the text or move to supplementary to make the main text a little bit more concise."*

Answer:

As already discussed in the answer to Reviewer 1, we made the following changes::

- Table 2, Figure 11 and 12 (in the first manuscript version) have been moved to Supplementary
- Figure 12 (14 in first manuscript version) has been slightly modified and moved to Supplementary.
- Sections 4.4.5 and 4.4.6 have been merged and reduced, part of the content has been moved to Supplementary. Figures 14 and 16 and Table 5 have been moved to Supplementary.

Despite the addition of some content in the introduction and methods section (following reviewers' suggestion), these changes allowed to reduce by 30% the number of figures, by 40 % the number of tables, and by 10% the total length of the paper.

The suggestion regarding revision of Figure 5 is discussed further below.

Specific comments:

*"Page 1 Line 13: example of ecological temp thresholds?"*

These thresholds are for example the 15°C for the PKD spread impacting salmonid fish. We changed this sentence to "ecological and economical temperature thresholds (spread of fish diseases and usage of water for industrial cooling)", linking it to our investigation of the legally imposed threshold of 25°C for water usage for industrial cooling in Switzerland (most importantly for nuclear power plants).

Review:

*"Page 2 Line 7: what is the global regime shift?"*

Answer:

Thanks for the question. It refers to a step change from the 1970s to the 1980s affecting climate and ecosystems mainly in central Europe but has been noticed also in other parts of the world. A good description can be found in Serra-Maluquer (2018):

"In the last four decades, a warming trend has been observed in the Iberian Peninsula; particularly, a rapid rise in temperatures has occurred since the 1980s followed by successive severe droughts in the 1990s, 2000s and 2010s (Gonzalez-Hidalgo and others, 2015). Such abrupt warming occurred in the transition from the 1970s to the 1980s, and it was partly linked to changes in the winter atmospheric circulation over the northern Atlantic Ocean (Hurrell, 1996) and impacted ecosystems worldwide by accelerating climate warming (Reid and others, 2016). This climate shift has led to warmer and more arid conditions on several European regions, generating harsher climatic conditions for beech forests".

The appropriate references have been added in the revised version.

Review:

*"Page 10 Line 7-8: why did the authors choose a 4 hours moving window average?"*

Answer:

The moving average is used to smooth the data avoiding to discard periods when the measured temperature was below 15°C just for one hour only. We also realize that the value of 4 hours was a typo, the actual value used is 3 hours (indeed moving window size for smoothing is always odd); this has been corrected in the revised version.

We made a sensitivity test on the value of the window size and on the total length of the periods while we developed this simple model. Results showed that the chosen values have little impact on the results (see Figures in the detailed answer to reviewer 2). This has been clarified in Section 3.4 (page 11 lines 19–20). However, we estimate that those plots (provided in answer to Reviewer 2) do not need to be included in the supplementary.

Review:

*"Figure 2: I like this figure but it is hard to tell which water station site is referring to which line on the top panel. Have the authors considered having the site labels point to the start or end of the 5 year moving average line for each site? This might improve the interpretation of site specific time series, but it also may make the figure too busy. Another option would be to order the site abbreviations in order of stream temperature from end of 5 year moving average line (i.e. year 2016) rather than what appears to be alphabetically-ordered currently."*
Answer:

Thank you for the suggestion. In our opinion, the best solution is ordering the site abbreviations in terms of stream temperature values at the end of the 5-year moving average period as the reviewer suggested, the Figure has been updated accordingly. This ordering is explained in the revised figure caption. Adding labels to the plot, or even only numbers, resulted in too much confusion and was not retained as a satisfying solution.

Review:

*"Figure 3. I think the labels could be ordered differently or point to the lines to which they correspond. See my comment for Figure 2 above. "*

Answer:

Absolutely. We modified it analog to Figure 2. We also realized that values for the Alte-Aare were missing in the plot (off-scale); we now divided these values by 4 to fit in the plot, and explained it in the caption of the figure. This high value for the specific discharge arises because the Alte Aare has a small proper catchment size (13 km$^2$), but has been artificially connected to the Aare. As a consequence, the discharge is far higher than the discharge we would expect from a catchment that small (see https://hydromaps.ch/#en/13/47.0536/7.2962/bl_hds).

Review:

*"Figure 5. Rather than plotting boxplots next to each other, I think plotting the water temperature trends vs. air temperature trends as well as discharge trends vs precipitation trends would convey more information. These scatter plots could also be colored by stream regime. You could keep the boxplots as marginal plots on the scatter plot figure to retain quartile and median information. It would be interesting to see when water and air temperature trends are correlated and when they are not."*

Answer:

Thanks for this suggestion. Our original version of this figure was exactly what you proposed, see Figures S20 and S21 in supplementary. As mentioned in the manuscript, there is far more noise in the water temperature trends than in the air temperature trends. Related to this, see details in our reply to Reviewer 1, there is non-negligible uncertainty around the trend values. For these reasons, we decided to present the results with boxplots, which implies a statistical preprocessing allowing to better visualize the signal in the noise. We prefer to keep the figures in the current version. However, Figures S20 and S21 have been added in the revised supplementary, and are discussed at the end of Section 4.2 (see page 16 lines 8–15).

When looking at S20, one could question the point in the top left corner. This point is a trend value for the water gauging station Rauss/Moutier, with meteorological values from Delemont. While the long-term trend in air temperature seems clear (0.43°C per decade for the period 1979-2018 and 0.46°C per decade for the period 1970-2018 obtained with the linear model for this station), no trend in air temperature is found for the last 20 years, explaining the unexpected position of this point. This is a good example of the noise obtained from a simple linear regression analysis (see more details about trend robustness on the answerer to Reviewer 1).

Review:

*"Page 13 into Page 14: "Indeed, for both pairs, the hypothesis of different mean is clearly rejected with p-values>0.15." Is this testing the difference between DLA and SPJ, and ALP and HYP? I assume so, but I think this could be written a little bit more clearly to make explicit."*

Answer:

Yes, here we refer to the values in Table 2, where each pair of regime trends is tested against each other to infer whether the means are similar or not. This has been clarified in the revised version (see page 15 lines 4–5).

Review:

*"Page 15 lines 12-13: It isn't clear to me how the authors concluded that air temperature is this main driver of water temperature trends for the SPJ catchments. Figure 5 shows a comparison between water and air temperature trends but all of the catchment types are grouped together in this figure so it is impossible to see the effect of air temp on water temp for SPJ catchments specifically. Please be clearer as to how you came to this conclusion. "*

Answer:

Indeed, the required Figure to illustrate this statement is for now missing from the paper. Figures corresponding to Figure 6, but for air temperature and precipitations, have been added in the supplementary (Figure S14 and S18 for the two time periods). By comparing the top-left panels of Figure 6 in the manuscript and of Figure S14, we clearly see that SPJ water temperature trends are, on average, close to the air temperature trends, which is not the case for HYP and ALP catchments (DLA catchments are discussed in Section 4.3). In Figures S20 and S21, we can also see that water temperature trends are spread around air temperature trends values for SPJ catchments, while they are systematically below for HYP and ALP.

Nevertheless, the paragraph on p.15 lines 12-16 has been modified to clearly state that we talk about the mean behavior of the trend and not about single catchments. In addition, comments related to the noise in the single trend comparisons have been added in this paragraph and in supplementary (Sections 4.2, page 16 lines 6–15, and S2.1). While part of this noise is caused by the method and the choice of the meteorological stations for the catchments, this noise shows that at the single catchment scale, independently of the regime, many factors other than the air temperature seem to be important. This is also mentioned in the revised version.

Review:

*"Page 15 line 20: include some citations for the statement that this is 'well known'"*

Answer:

We can cite for instance Råman Vinnå (2018) and Webb (2007). These references have been added in the revised manuscript (page 17 line 5).

Review:

*"Figure 7: I suggest adding the label 'inflow' and 'outflow' to the figure itself to help the reader quickly understand the figure rather than having to read through the legend to understand which line is inflow and which is outflow."*

Answer:

This has been modified as suggested in the revised version. Indeed, this eases reading of the figure. Additionally, to avoid confusion, we changed the colors in the bottom panel since the meteorological stations do not necessarily match with the rivers shown (the number of available water temperature and meteorological stations can differ).

Review:

*"Page 18 line 4: Is 'intra-annual' not 'infra-annual' more appropriate here?"*

Answer:

Good catch. Indeed, this is a language mistake, "infra" meaning "below" while "intra" means "within". 'Infra-annual' has been replaced by 'intra-annual' in the whole text. Thanks for pointing this out.

Review:

*"Figure 8: indicate what the panel months mean (DJF, JJA, etc...). I was confused until read the main text."*

Answer:

The letters refer to the initials of the three months of a season defined as December-January-February (DJF) and so on. This information has been added in the caption of the figure.

We thank again the reviewers for the useful and pertinent comments and for taking the time to go through this comprehensive manuscript. These comments led to a substantial improvement of the manuscript and we hope that the revised version answers to the raised questions, comments and suggestions.

Adrien Michel, on behalf of all authors.

New figure

[Figure]

**Figure S9.** Robust trends plotted against simple linear regression trends for the period 1999-2018 for water temperature (top-left), air temperature (top-right), and discharge (bottom-left). The square indicates the mean value and the RMSE is indicated in °C (top) or in % (bottom).

**New figure**

[Figure]

**Figure S10.** Robust trends plotted against simple linear regression trends for the period 1979-2018 for water temperature (top-left), air temperature (top-right), and discharge (bottom-left). The square indicates the mean value and the RMSE is indicated in °C (top) or in % (bottom).

**New figure**

**Period 1999–2018, normal linear regression**

**Figure S11.** Trends for the period 2000-2018 (red) and 1999-2017 (green) plotted against trends for the period 1999-2018 for water temperature (top-left), air temperature (top-right), discharge (bottom-left) and precipitation (bottom-right). The square indicates the mean value and the RMSE is indicated in °C (top) or in % (bottom).

**New figure**

[Figure]

**Figure S12.** Trends for the period 1980-2018 (red) and 1979-2017 (green) plotted against trends for the period 1979-2018 for water temperature (top-left), air temperature (top-right), discharge (bottom-left) and precipitation (bottom-right). The square indicates the mean value and the RMSE is indicated in °C (top) or in % (bottom).

**S2   Supplementary material about results**

**S2.1    Long-term and trend analysis**

This Section presents additional results for Sections 4.1 and 4.2 of the main article. Figure S13 shows the decadal mean of air temperature anomaly (similar to Figure 2 bottom panel and Figure 4 in the main text), Table S3 presents the results of the two-sided Wilcoxon test used to assess whether differences between regimes are significant in terms of temperature trends, and Figure S14 shows the air temperature and precipitation trends for the four different regimes, and classified upon area, elevation and glacier-covered fraction, as Figure 6 in the main text.

[revised manuscript text omitted]

**New figure**

[Figure]

**Figure S14.** Air temperature and precipitation trends for the period 1999-2019. Top left: classified according to the four different hydrological regimes (DLA = downstream lake regimes, ALP = alpine regimes, SPJ = Swiss Plateau/Jura regimes and HYP = strong influence from hydropeaking). Top right: classified according to catchment area. Bottom left: classified according to the catchment elevation. Bottom right: classified according to the glacier coverage.

[Figure]

**Figure S15.** Relative discharge and precipitation decadal means of anomalies with respect to the 1920-2018 average for 20 catchments and 22 MeteoSwiss homogeneous stations with data available since 1920 (upper two plots). Yearly mean of the NAO and AMO (lower two plots).

[Figure]

**Figure S16.** Distributions of trends of water and air temperature (left) , and normalized discharge and normalized precipitation (right), for the periods 1979-2018 for the 27 catchments where data are available for temperature and discharge (see Table 1 in main text).

Updated figure

[Figure]

**Figure S17.** Water temperature and discharge trends for the period 1979-2019 for the 27 catchments where data are available for water temperature and discharge (see Table 1 in main text). Top left: classified according to the four different hydrological regimes (DLA = downstream lake regimes, ALP = alpine regimes, SPJ = Swiss Plateau/Jura regimes and HYP = strong influence from hydropeaking). Top right:  classified according to catchment area. Bottom left:  classified according to the catchment elevation. Bottom right:  classified according to the glacier coverage.

**New figure**

[Figure]

**Figure S18.** Air temperature and precipitation trends for the period 1979-2019 for the 27 catchments where data are available for water temperature and discharge (see Table 1 in main text). Top left: classified according to the four different hydrological regimes (DLA = downstream lake regimes, ALP = alpine regimes, SPJ = Swiss Plateau/Jura regimes and HYP = strong influence from hydropeaking). Top right: classified according to catchment area. Bottom left: classified according to the catchment elevation. Bottom right: classified according to the glacier coverage.

[Figure]

**Figure S19.** Left: Distribution of catchment area for four different regimes (DLA = downstream lake regimes, ALP = alpine regimes, SPJ = Swiss Pateau/Jura regimes and HYP = strong influence from hydropeaking). Right: Temperature trends for SPJ regime catchments only.

**New figure**

[Figure]

**Figure S20.** Water temperature trends plotted against air temperature trends. Values are colored by catchment area (top-left), glacier covered catchment fraction (top-right), mean catchment elevation (bottom-left) and regimes (bottom-right). Period 1999-2018.

**New figure**

[Figure]

**Figure S21.** Water temperature trends plotted against air temperature trends. Values are colored by catchment area (top-left), glacier covered catchment fraction (top-right), mean catchment elevation (bottom-left) and regimes (bottom-right). Period 1979-2018.

**S2.2 Lake effect**

This Section presents plots for the four lakes not shown in the main text Section 4.3: Lake Walen (Figure S22), Lake Luzern (Figure S23), Lakes Brienz and Thun (Figure S24), and Lake Biel (Figure S25). The values for the various trends presented are shown in Table 3 in the main text.

[Figure]

**Figure S22.** Lake Walen: Water temperature anomalies and trends for inflow and outlet stations (top), air temperature anomalies and trends for surrounding MeteoSwiss stations (bottom). The period for trend  computation is 1979-2018. The abbreviation for water gauging stations and for MeteoSwiss stations are given in Table 1 in main text and in Table S2.

[Figure]

**Figure S23.** Lake Luzern: Water temperature anomalies and trends for inflow and outlet stations (top), air temperature trends for surrounding MeteoSwiss stations (bottom). The period for trend  computation is 1979-2018, except for the Engelberger Aa in Buochs (Eaa-Buo) where the trend is computed over the period 1999-2018. The abbreviation for water gauging stations and for MeteoSwiss stations are given in Table 1 in main text and in Table S2.

[Figure]

**Figure S24.** Lakes Brienz and Thun: Water temperature anomalies and trends for inflow and outlet stations (top), air temperature anomalies and trends for surrounding MeteoSwiss stations (bottom). The period for trend  computation is 1979-2018. The abbreviation for water gauging stations and for MeteoSwiss stations are given in Table 1 in main text and in Table S2.

[Figure]

**Figure S25.** Lake Biel: Water temperature anomalies and trends for inflow and outlet stations (top), air temperature anomalies trends for surrounding MeteoSwiss stations (bottom). The period for trend  computation is 1979-2018. The abbreviation for water gauging stations and for MeteoSwiss stations are given in Table 1 in main text and in Table S2.

**S2.3   Seasonal trends and relation with air temperature and precipitation**

This Section presents additional results related to Section 4 of the main text. Figures S26 and S27 show the  decadal evolution of air temperature and precipitation for the four seasons, similar to Figures 8 and 9  in the main text for stream temperature and discharge.

Table S6 shows the correlation between trends of various variables. As discussed in the main text, these correlations are mostly not significant and thus not considered in the study.

Figures S28, S29 and S30 show the yearly anomalies in stream temperature, discharge, air temperature and precipitation in winter and fall, similar to Figures  11 in the main text which  presents summer.

 Figure S31 shows the snow water equivalent (SWE) at the beginning of various months over the whole country, Figure S32 the evolution of spring melt evolution, obtained by subtracting first of June to first of March SWE, and Figure S33 shows the evolution of the summer mass balance for  7 Swiss glaciers.

Finally, Figure S34 and Table S7 show additional content for Section 4.4.5 of the main text. Figure S34 shows the annual difference between summer and winter means for all catchments with data since at least 1980. A 5-year moving average window is applied for noise reduction. Note the ~14 years cycle present in the data with an amplitude of about 0.5 °C, probably caused by large scale atmospheric phenomena (as also found in Webb and Nobilis (2007)). The year-to-year variations of the temperature difference anomaly are more driven by this oscillation than by the underlying trend. Nevertheless, there is a clear evolution on the intra-annual variability: the computed trend indicates an increase of 0.3±0.1 °C per decade, which corresponds to a change of +1.2 °C over the studied period. The mean raw intra-annual variability equals 9.8 °C, with a standard deviation of 3.6 °C. This represents an increase of 10% to 20% of the variability for individual catchments.

Table S7 shows the correlations between water temperature and water temperature from previous seasons, between discharge and precipitation from previous seasons, and between water temperature and precipitation from previous. They were obtained with the same method as the data in Table 3 in main text. This Table shows that for water temperature there is almost no correlation and calculated values are mostly not significant. The only observed signal is from one season directly to the next one, but it is far weaker and less significant than the correlation with air temperature during the season (see Table 3 in main text). There is also no strong correlation between precipitation and discharge more than one season apart. The correlation with the next season is weak and significant only for a few catchments, showing that the groundwater storage plays an important buffer role. A weak correlation is also seen between winter and the following summer, showing the influence of the remaining snow in summer for a few catchments. Regarding correlation between precipitation and water temperature, only two values are significant for more than 10 catchments. There is a negative correlation between spring precipitation and summer stream temperature, which is discussed in the main text. There is also a positive significant correlation for 15 catchments from spring to the following year spring, but since no real physical process was found to explain it, it is assumed to be noise in the results.

[Figure]

**Figure S26.** Air temperature seasonal anomalies  for the 14 catchments where data are available since 1970 (see Table S2). Anomalies with respect to the 1970-2018 period.

[Figure]

**Figure S27.** Precipitation seasonal relative anomalies  for the 26 stations where data are available since 1960 (see Table S2). Anomalies with respect to 1960-2018 period.

Finally, Figure S35 shows some additional details about alpine catchments discussed in Section 4.4.4 in the main text and Figures S36 and S37 show plots similar to Figure 15 in the main text but for the Arve in Geneva and for the Lütschine in Gsteig.

**Table S6.** Correlation between the trends of water and air temperature (left), water temperature and discharge (middle) and discharge and precipitation (right). Correlations are computed between annual and seasonal trends, and by taking one value per catchment and constructing ordered vectors of values. The number in parenthesis indicates the p-value of the null-hypothesis (no correlation). Since the computation here is different  from the one in Table 4 in main text (where correlation is computed from full time series and then averaged between catchment), the two tables cannot be compared.

| Water and air temperature trends | | Water temperature and discharge trends | | Discharge and precipitation trends | |
|---|---|---|---|---|---|
| **Period** | **cor.** | **Period** | **Cor.** | **Period** | **Cor.** |
| Yearly | -0.18 (0.19) | Yearly | -0.25 (0.08) | yearly | 0.08 (0.01) |
| Winter | -0.13 (0.36) | Winter | -0.50 (<0.01) | Winter | 0.36 (0.02) |
| Spring | 0.02 (0.87) | Spring | -0.35 (0.01) | Spring | 0.11 (0.43) |
| Summer | -0.09 (0.51) | Summer | -0.05 (0.72) | Summer | 0.33 (0.01) |
| Fall | -0.26 (0.07) | Fall | -0.26 (0.06) | Fall | 0.34 (0.58) |

[Figure]

**Figure S28.** Winter anomalies in stream temperature, air temperature, relative discharge and relative precipitation for all catchments. Anomalies are computed with respect to the 1999-2018 mean for each catchment.

[Figure]

**Figure S29.**  Spring anomalies in water temperature,  air temperature, relative discharge and relative precipitation for all 52 catchments. Anomalies are computed with respect to the 1999-2018 mean for each catchment.

[Figure]

**Figure S30.** Fall anomalies in stream temperature, air temperature, relative discharge and relative precipitation for all catchments. Anomalies are computed with respect to the 1999-2018 mean for each catchment.

[Figure]

**Figure S31.** Snow water equivalent in spring over the entire Switzerland at the beginning of the months, from March to July. Obtained from Magnusson et al. (2014) and provided by the WSL Institute for Snow and Avalanche Research (SLF).

[Figure]

**Figure S32.** Snow melt evolution in spring over the entire Switzerland, obtained by subtracting first of June to first of March SWE.

[Figure]

**Figure S33.** Summer mass balance for 9 7 Swiss glaciers, from (GLAMOS, 2018).

[revised manuscript text omitted]